# Cytokine gene polymorphism and parasite susceptibility in free-living rodents: Importance of non-coding variants

Agnieszka Kloch[1]*, Ewa J. Mierzejewska[2], Renata Welc-Falęciak[3], Anna Bajer[4], Aleksandra Biedrzycka[5]

**1** Department of Ecology, Faculty of Biology, University of Warsaw, Warszawa, Poland, **2** Wild Urban Evolution and Ecology Laboratory, Centre of New Technologies, University of Warsaw, Warszawa, Poland, **3** Department of Parasitology, Faculty of Biology, University of Warsaw, Warszawa, Poland, **4** Department of Eco-epidemiology of Parasitic Diseases, Faculty of Biology, University of Warsaw, Warszawa, Poland, **5** Institute of Nature Conservation, Polish Academy of Sciences, Kraków, Poland

* a.kloch@uw.edu.pl

## Abstract

Associations between genetic variants and susceptibility to infections have long been studied in free-living hosts so as to infer the contemporary evolutionary forces that shape the genetic polymorphisms of immunity genes. Despite extensive studies of proteins interacting with pathogen-derived ligands, such as MHC (major histocompatilbility complex) or TLR (Toll-like receptors), little is known about the efferent arm of the immune system. Cytokines are signalling molecules that trigger and modulate the immune response, acting as a crucial link between innate and adaptive immunity. In the present study we investigated how genetic variation in cytokines in bank voles *Myodes glareolus* affects their susceptibility to infection by parasites (nematodes: *Aspiculuris tianjensis*, *Heligmosomum mixtum*, *Heligmosomoides glareoli*) and microparasites (*Cryptosporidium sp*, *Babesia microti*, *Bartonella sp.*). We focused on three cytokines: tumour necrosis factor (*TNF*), lymphotoxin alpha (*LTα*), and interferon beta (*IFNβ1*). Overall, we identified four single nucleotide polymorphisms (SNPs) associated with susceptibility to nematodes: two located in LTα and two in IFNβ1. One of those variants was synonymous, another located in an intron. Each SNP associated with parasite load was located in or next to a codon under selection, three codons displayed signatures of positive selection, and one of purifying selection. Our results indicate that cytokines are prone to parasite-driven selection and that non-coding variants, although commonly disregarded in studies of the genetic background of host-parasite co-evolution, may play a role in susceptibility to infections in wild systems.

## 1. Introduction

Parasite-driven selection is considered a key factor shaping the evolution of the components of the immune system. In vertebrates, the immune system comprises many interacting molecules, yet the mechanisms of this selection have been comprehensively studied only in the case

**Data Availability Statement:** The raw reads from Illumina sequencing are available from SRA archive PRJNA395763. Minimal data set underlying the results containing individual genotypes and data on

on individual parasite load, body mass, sex, and sampling details are stored in Open Science Framework repository https://doi.org/10.17605/OSF.IO/QKFUM.

**Funding:** The work was supported by grant no. 2012/07/B/NZ8/00058 from the Polish National Science Centre to AK. The funders had no role in study design, data collection and analysis, decision to publish, or preparation of the manuscript.

**Competing interests:** he authors have declared that no competing interests exist.

of major histocompatibility complex (MHC) genes e.g. [1]. Exceptionally high variation of MHC genes is contributed to parasite-driven balancing selection which maintain alleles within a population at higher frequencies than expected. Balancing selection operates through several mutually not exclusive mechanisms. A heterozygote advantage occurs when heterozygotes are able to recognise a wider array of pathogen-derived motifs compared to homozygotic individuals [2]. A rare-allele advantage occurs when parasites are most likely to adapt to the most frequent host genotypes, thus rare variants are usually beneficial [3]. Those variants are then favoured by natural selection and their frequency increases until they become common and targeted by parasite counter-adaptations. Such heterogeneity over time (and space) results in the third mechanisms, the fluctuating selection [4, 5]. Though the MHC plays an important role in antigen-based pathogen recognition, it is not the only nor the main factor responsible for resistance against pathogens [6, 7]. Recently, researchers have focused on the components of innate immunity, in particular on the toll-like receptors eg. [8–10], but the evolution of other elements of the immune system, including cytokines, remains poorly understood.

Cytokines, signalling molecules capable of triggering and modulating the immune response, are the crucial link between innate and adaptive immunity. As an efferent arm of the immune system they are expected to be more evolutionarily constrained than elements of the afferent arm [11]. Nonetheless, a handful of studies have so far reported signatures of balancing or positive selection in this group of molecules. For instance, balancing selection was found in genes coding for interleukins *Il-1B*, *Il-2* and tumour necrosis factor *TNF* in field voles [12], and in interleukin *Il-10* and protein *CD14* in humans [13]. Another kind of evidence in support of contemporary parasite-driven selection operating on cytokines comes from significant associations between genetic variants and susceptibility to infections. In humans, polymorphism within the lymphotoxin alpha (*LTα*) has been linked to several diseases, such as leprosy [14] and malaria [15]. Among rodents, both positive and negative associations between variation in interleukins and infections with uni- and multicellular parasites have been found in the field vole [16]. Only one study focusing on non-coding variance in a free-living mammal revealed that variants affecting susceptibility may be located outside the coding part of a cytokine gene [17].

In the present study we investigated a system including the bank vole *Myodes glareolus* and its parasites from three locations in NE Poland, differing in the composition of the community of gastrointestinal helminths [18–21]. There are three dominant nematode species in this study system–the oxyurid *Aspiculuris tianjinensis* (formerly *A. tetraptera*) and two Heligmosomoidae worms that show site-specific patterns: *Heligmosomum mixtum* is absent at the site Pilchy, while *Heligmosomoidae glareoli* is rare at the site Urwitałt. Apart from nematodes, voles are also infected with cestodes and a range of vector-transmitted blood parasites and protozoa e.g. [18, 22]. Previous studies have shown that the presence of a blood parasite decreases winter survival in root voles [23], and that nematode infections reduce reproductive fitness [24] and depauperate gut microbiota diversity [25] in mice. In our study system, we have previously reported associations between variation in the MHC and TLR (Toll-like receptor) genes [10, 21] and susceptibility to infections. Both gene families code for proteins directly interacting with structures derived from pathogens. TLRs recognize general pathogen-associated motifs triggering the innate immune response, and MHC present specific antigens as a part of the adaptive immune response. Here, we completed this picture by focusing on cytokines, which act as a functional link between those two branches of the immune system.

We selected three genes coding for non-interleukin cytokines, as this group is relatively less frequent studied than interleukins (e.g. [16]): tumour necrosis factor (*TNF*), lymphotoxin alpha (*LTα*) and interferon beta (*IFNβ1*). Their versatile role within the immune system, along with evidence from human studies demonstrating their importance in responding against

various pathogens, makes them promising candidate genes for studying the mechanisms of parasite-driven selection.

*TNF* and lymphotoxin alpha (*LTα*) genes have similar structure, and until recently *LTα* was called tumour necrosis factor beta (*TNFβ*). The genes, separated by just 1600bp, are located in the chromosome 17 in the proximity of the genes encoding major histocompatibility complex (MHC), key component of the adaptive immune system. Tumour necrosis factor (*TNF*) initiates the acute phase response and acts as an endogenous pyrogen contributing to inflammation. By stimulating the endothelial cells in blood vessels it plays a role in preventing the pathogen from entering the bloodstream and in locally containing an infection [26]. It has been shown that genetic variants located in the promoter sequence of TNF affect resistance against viral infections in bank voles [17]. Here, we expanded this study by examining associations between polymorphisms within TNF and susceptibility to other groups of pathogens, namely nematodes and blood microparasites.

*LTα* is expressed in lymphocytes and plays a major role in immunomodulation and signal transduction within the immune system. It induces inflammation and is the key factor facilitating the innate immune response through activation of IFNβ and NF-κB pathways [14, 27] *LTα* is necessary for effective adaptive responses involving T and B lymphocytes [28] and for developing intestinal lymph nodes [29]. Thus, we expected to confirm links between *LTα* variants and resistance against gastrointestinal nematode infections. Human GWAS studies reported associations of single-nucleotide polymorphisms, including promoter and intron variants of TNF and *LTα* with susceptibility to several bacterial and protist infections, including *Mycobacterium leprae*, *M. tuberculosis* and *Plasmodium falciparum* (reviewed in [30]). Thus, we expected to find associations between polymorphisms in those genes and resistance against microbial pathogens. *IFNβ1* is produced in response to viral but also bacterial infections [31], and through a specific pathway, it plays a major role in linking innate and adaptive immunity. This cytokine has not yet been studied in the context of parasite-driven selection, and we expected that variation within *IFNβ1* gene may play a role in resistance against bacterial pathogens.

Helminth infections, which are prevailing pathogens in wild animals, show immunomodulatory effects supressing acute, inflammatory Th-1 type reactions, and it has been shown that presence of nematodes may alter the outcome of bacterial infections in free-living mammals [32]. Assuming that polymorphisms within inflammatory cytokines may play a role in this process, we expected to find associations between studied gene variants and presence of helminth infections.

## 2. Material and methods

### 2.1. Samples and parasite screening

In the present work we used samples collected in autumn (September) of 2005 and in 2016 at three sites in NE Poland: Urwitałt (53.80004 N, 21.65250 E), Pilchy (53.70268N, 21.80116 E), and Tałty (53.89362N, 21.55392 E). All sites are located in coniferous and mixed forests, situated within ~20 km distance. The sites Urwitałt and Pilchy lie on opposite banks of Lake Śniardwy, and Tałty is ~10km north of Urwitałt. All three sites are on public ground managed by the Polish State Forests and no specific permission to access the land was required. The number of samples collected in each year at each site is given in Table 1.

The field procedures followed the guidelines of the National Ethics Committee for Experimentation on Animals and were approved by the Local Ethical Committee no. 1 in Warsaw, decisions no. 280/2003 and 304/2012. Parasite screening followed protocols described in [18, 21]. In brief, voles were live-trapped in wooden traps and transported to the field station,

**Table 1. Number of samples analysed per gene, site and year.** As explained in the Methods, only samples with all SNP successfully genotyped were included in the association analysis, these are presented in the bottom part of the table. For technical reasons, not all individuals were genotyped in all genes.

| | Sequenced | | | | | |
|---|---|---|---|---|---|---|
| | Pilchy | Tałty | | Urwitałt | | total |
| | 2005 | 2005 | 2016 | 2005 | 2016 | |
| TNF | 30 | 18 | 8 | 13 | 11 | 80 |
| LTα | 45 | 17 | 8 | 44 | 12 | 126 |
| IFNβ1 | 41 | 0 | 0 | 44 | 0 | 85 |
| | After quality filtering | | | | | |
| TNF | 23 | 18 | 7 | 8 | 11 | 67 |
| LTα | 38 | 17 | 8 | 40 | 11 | 114 |
| IFNβ1 | 41 | 0 | 0 | 44 | 0 | 85 |

*where they were anesthetized* with isoflurane and killed by cervical dislocation. GI helminths were identified in autopsies. Worms counts were considered as the parasite abundance. Infections with intestinal protist *Cryptosporidium* sp. were identified in faecal smears using the Ziehl-Nielsen staining technique [33]. Blood pathogen *Bartonella sp.* was identified by PCR using degenerate primers described by [34], and for *Babesia microti* we used the protocol described by [35]. Details of the PCR reactions are given in S1 Table.

## 2.2. Sequencing and genotyping

DNA was extracted from vole ears using the Qiagen DNeasy Blood & Tissue Kit. To amplify partial sequences of *TNF*, we used primers designed by [12]; *LTα* and *IFNβ1* were amplified using degenerate primers designed in our group (S1 Table). To minimize PCR errors, we used high fidelity polymerase Phusion or Q5 (NewEngland Biolabs). The mix contained 10uM dTNPs, 0.5uM of each primer, 0.02U/ul of the polymerase, and 10–50 ng of genomic DNA. PCR conditions specific for each locus are given in S1 Table. Amplified regions spanned most of the expressed exons and respective separating introns (S2 Table).

Samples from 2005 and 2016 were processed separately following the same protocol. In both experiments, the amplicons were pooled for each individual as described previously [10] and purified twice using CleanUp kit (Aabiot). The libraries were constructed using Nextera XT DNA Library Preparation Kit, and sequenced using MiSeq Reagent Kit v3 in an Illumina MiSeq machine. The only difference between experiments was the number of cycles used for sequencing: samples from 2005 were sequenced using 2x75 paired-end kit, and samples from 2016 were processed using 2x150bp paired-end kit. Both runs gave high coverage exceeding 1000x per site and per sample.

Reads from both runs were processed in the same pipeline as described previously [10]. In brief, adapter-clipped reads were mapped using bwa-mem ver. 0.7.12 with default parameters [36] against a reference constructed from regions including genes of interests extracted from bank vole genome (PRJNA290429). Duplicated read-pairs were removed using Picard Mark-Duplicates (http://picard.sourceforge.net), and variants were called in two-step procedure in FreeBayes v 1.1.0–60 [37]. In the first round, potential variants were called using the following parameters: minimal fraction of alternate allele of 20% as recommended by [38], minimum number of reads supporting alternate allele > 2, and minimal read coverage > 5. The results were filtered using vcffilter v. 41 (https://github.com/ekg/vcflib) with conservative criteria: remove low quality calls (QUAL/AO > 10), remove loci with low read depth (DP > 10), remove alleles present in only one strand (SAF > 0 & SAR > 0), remove alleles that are only observed by reads placed to the left or right (RPR > 0 & RPL > 0). The resulting in high

confidence variants were used to construct haplotypes in a second round of variant calling in FreeBayes where physical position on a read was used for phasing by specifying —max-complex-gap 37. A fraction of SNPs (5 of 36 in *TNF*, and 6 of 18 in *LTα*) could not be phased by FreeBayes; these were computationally assigned to DNA strands using PHASE algorithm [39]. Reconstructed alleles were converted to fasta format using vcfx [40, 41]. The reading frames and intron/exon structure were resolved through an alignment to the orthologue mouse sequences (*TNF* Y00467.1, *LTα*: Y00137.1, *IFNβ1* NP_034640).

Sequenced parts of *TNF* and *LTα* spanned ca. 800bp and each contained 3 exons. *IFNβ1* gene is intronless. The number of polymorphic sites per locus was similar considering the length of the sequenced fragment (S2 Table). However, most variance in TNF was located in introns, whereas in exons we found only 2 SNPs forming 3 haplotypes. After filtering out variants in linkage equilibrium, not in Hardy-Weinberg equilibrium, with minor allele frequency <5% and present in fewer than 5 hosts, of the initial 18 SNPs in *TNF*, 17 in *LTα*, and 8 in *IFNβ1*, we retained one SNP in *TNF*, 7 SNPs in *LTα*, and two in *IFNβ1*.

## 2.3. Associations between genetic variants and parasite load

The sample size was strongly limited by ethical considerations. Unable to sample more animals to test for the effect of rare alleles, we filtered the data to remove variables that were too few to produce robust results [42]. First, we removed variants and individuals with missing SNPs, keeping only those with a genotyping rate of 100%. Next, SNPs were filtered in PLINK v1.90 [43] so that only those in Hardy-Weinberg equilibrium (threshold of p<0.001) and in linkage equilibrium (r>0.7) were retained. (We tested for LD values from r>0.6 to r>0.9 and all the values resulted in the same set of SNPs.) This ensured that genetic variants fitted in the models can be treated as independent explanatory variables.

In a second step, we excluded from the models variants with minor allele frequency (MAF) <5% and those present in fewer than 5 animals (see S3 Table), as we lacked statistical power to test for their effects. The number of samples before and after filtering in given in Table 1. Although new beneficial mutations may initially be present in one or few individuals, in practice it is impossible to test for effect of alleles of MAF <5% when the relative risk of disease is lower than 1.5 [44]. Thus, due to computational limitations, our analysis omitted the effects of rare alleles.

Among the parasites found in voles, there were five nematode species, one cestode, one intestinal protist and two blood parasites. The prevalence varied from 5 to 60% but due to limitations of generalized linear models, we run only models where prevalence was 20–80% for a given parasite/ gene combination (S4 Table, S1 Fig). In total, we constructed 23 models, 16 with infection presence/absence as reponse variable, and 7 with parasite abundance as response variable. We tested for effect of *TNF* variants on prevalence and abundance of *A. tjaniensis*, *H. mixtum* and prevalence of *Cryptosporidium*, *Babesia microti* and *Bartonella sp*, *LTα* variants on prevalence and abundance of *A. tjaniensis*, *H. mixtum* and prevalence of *Cryptosporidium*, *Babesia microti* and *Bartonella sp*, and *IFNβ1* variants on prevalence and abundance of *A. tjaniensis*, *H. mixtum*, *H. glareoli* and prevalence of *Cryptosporidium*, *Babesia microti and Bartonella sp*.

We adapted a two-step procedure used in several association studies of free-living mammals where minimal model with non-genetic terms is fitted first, and then the effects of genetic terms are tested eg. [16, 45]. Such a procedure prevents overfitting the model. The non-genetic terms were: year and site of sampling, host sex, and host age approximated by its body mass (S5 Table). If p<0.1, these terms were included in the models as co-factors. As response variables, we used either i) parasite presence/absence, or ii) parasite abundance (number of

parasites of a given species per host). Parasite presence/absence was modelled using binomial distribution with logit link function, and abundance was fitted using Poisson distribution and log link function in R [46]. To control for overdispersion, in abundance models we used quasi-Poisson errors and in presence models quasi-binomial errors implemented in the glm function in the R library {stat}. The effect sizes were estimated using partial R2 implemented in the R package {rsq}. The significance of terms was determined using LR type III tests.

After fitting minimal non-genetic models including the non-genetic terms, we tested for genetic effects. The models were constructed as above, with parasite presence/absence or abundance as response variables. The explanatory variables were: significant non-genetic terms and genotypes in each SNP retained after filtering. Number of successfully genotyped individuals differed between studied genes, so we constructed separate sets of models for each studied gene. To control for multiple comparisons we applied conservative Bonferroni correction. In total, we tested for the effect of 10 SNPs retained after filtering (one in *TNF*, seven in *LTα*, and two in *IFNβ1*), so as a significance threshold corresponding to $\alpha = 0.05$ we adapted $\alpha = 0.05/10 = 0.005$.

### 2.4. Tests of selection

To identify potential targets of selection in exonic regions, we computed phylogenetically controlled codon-based tests which are suitable for identifying sites under selection using sets of sequences from a single species [47]. For the calculations, we used DataMonkey server [48]. Prior to analysis, we tested for possible recombinations using GARD, Genetic Algorithm Recombination Detection [49]. We computed three models based on dN/dS ratio: MEME (Mixed Effects Model of Evolution), which employs mixed-effects maximum likelihood to infer synonymous and nonsynonymous substitution rates and detect episodic positive selection under a proportion of branches [50]; FUBAR (Fast Unconstrained Bayesian Approximation), which estimates the dN/dS ratio using a Bayesian approach to detect sites under pervasive diversifying selection [51]; and FEL (Fixed Effects Likelihood), which uses a maximum-likelihood approach to detect pervasive diversifying selection using corresponding phylogeny [47]. Contrary to the other two tests, FEL assumes constant selection pressure across phylogeny.

## 3. Results

### 3.1. Cytokine polymorphisms and susceptibility to infections

After filtering the variants, in the models we included one SNP in *TNF* gene, 7 SNPs in *LTα* and two in *IFN*. Contrary to expectations, in either gene we found no association between genetic variants and susceptibility to infections with microparasites: intestinal protozan *Cryptosporidium*, bacteria *Bartonella sp.*, and Apicomplexan blood parasite *Babesia microtii*.

Moreover, there were no significant associations between TNF variants and susceptiblity to helminths (S6 Table).

None of the LTa variants affected parasite prevalence but after correcting for multiple comparisons variant *LTα* 535 turned out to be significantly associated with abundance of the nematode *Heligmosomum mixtum*. The SNP *LTα* 522 was located in an intron, and homozygotes TT were infected by more worms (3.03 on average) compared to genotypes GG and TG (1.25 and 0.94 respectively, Fig 1).

SNPs in IFNβ affected risk of infection and parasite load with the nematode *H. glareoli*. Voles heterozygous in *IFNβ1* 105 were less frequently infected with the nematode *H. glareoli* (11.7%) compared to homozygotes TT (57.14%) and CC (18.9%, Fig 1, Table 2), and they also harboured fewer *H. glareoli* worms (0.411) compared to 1.78 vs 1.19 in the respective

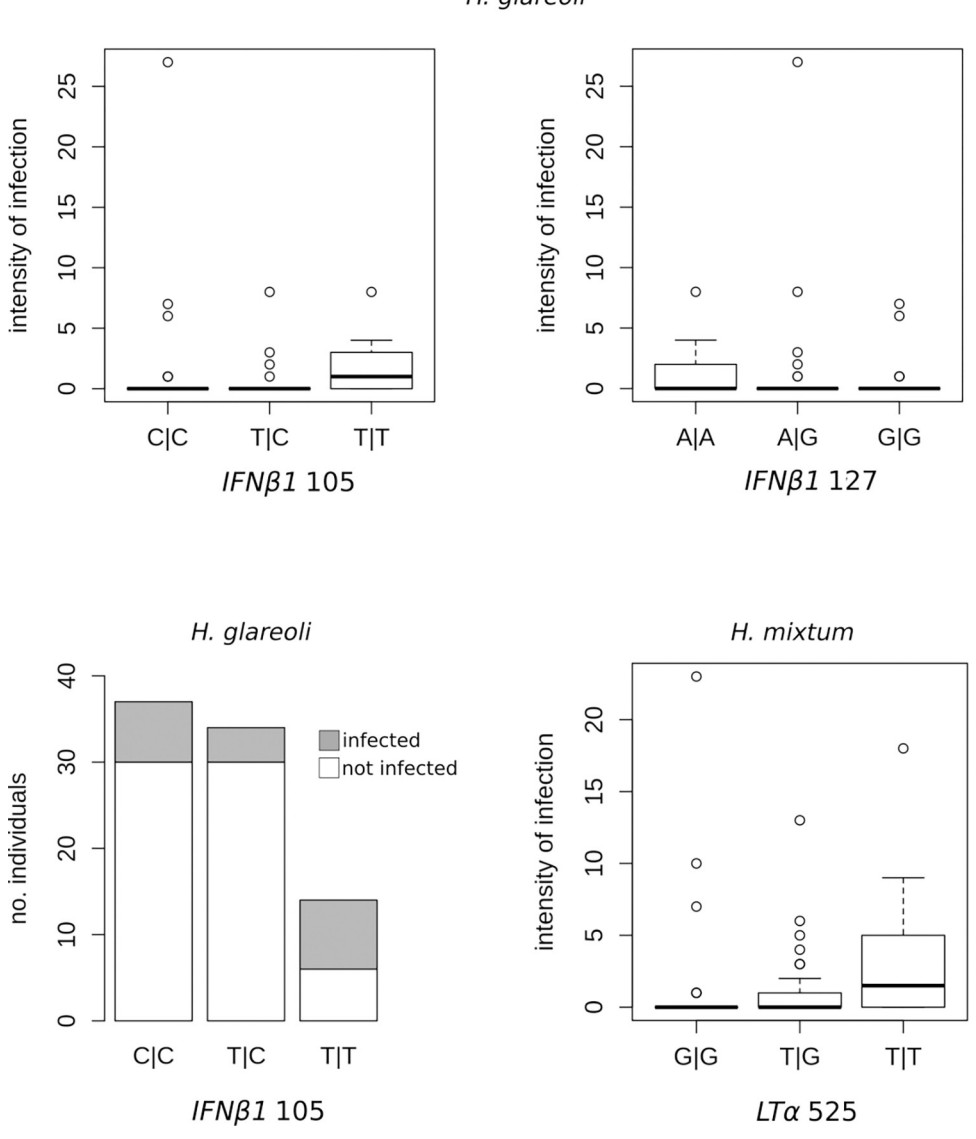

**Fig 1. Effect of SNP genotypes in LTα and IFNβ1 on the risk of infection and parasite abundance.** Infection intensity is expressed as the number of worms per host); the top and bottom of a box are 25 and 75% percentiles, the outliers are points 1.5 times the interquartile range above the third quartile.

homozygotes. In *IFNβ1* 127 homozygotes AA were most often infected (38%) compared to 18.2% of infected individuals with the genotype AG and 16.1% with GG. *IFNβ1* 105 coded for synonymous substitution, and *IFNβ1* 127 coded non-synonymous substitution Asn→ Ser.

### 3.2. Signatures of selection

No signal of recombination was detected in any of the studied genes using the GARD method. MEME did not detect any codon thus we did not find evidence for episodic selection. FEL and FUBAR results were generally consistent across loci (S7 Table, Fig 2). In TNF gene, two codons located in the exon 4 bore signatures of purifying selection, as indicated by FUBAR only. FUBAR and FEL detected two sites under purifying selection in *LTα*. Two codons in this gene were under positive selection according to FUBAR, one of those codons comprised SNP 371

**Table 2. Summary of GLM models showing significant effect of SNP genotype at given locus on the parasite load.** Table presents summary of GLM models including non-genetic terms as indicated in S4 Table. All models are given in S6 Table. $R^2$ is partial coefficient of determination (effect size), $\chi^2$ and p-values are based on LR type III test. To control for multiple comparisons when testing for the effect of several genetic variants, we used conservative Bonferroni correction. For 10 genetic terms (SNPs) tested, the critical p-level corresponding to $\alpha$ = 0.05 is 0.005. p-values of genetic terms significant after correction are given in bold.

| gene | nematode | variable | $R^2$ | $\chi^2$ | df | p |
|------|----------|----------|-------|----------|-----|---|
| | | **Presence / absence** | | | | |
| IFNβ1 | H. glareoli | IFNβ1 105 | 0.267 | 30.025 | 2 | **$3.21 \times 10^{-7}$** |
| | | IFNβ1 127 | 0.077 | 7.636 | 2 | 0.022 |
| | | site | 0.492 | 91.094 | 1 | <0.001 |
| | | host body mass | 0.084 | 7.149 | 1 | 0.007 |
| | | host sex | 0.191 | 17.498 | 1 | <0.001 |
| | | **Abundance** | | | | |
| IFNβ1 | H. glareoli | IFNβ1 105 | 0.136 | 12.986 | 2 | **0.00151** |
| | | IFNβ1 127 | 0.172 | 12.546 | 2 | **0.00189** |
| | | site | 0.180 | 42.039 | 1 | <0.001 |
| | | host sex | 0.127 | 10.558 | 1 | 0.001 |
| LTα | H. mixtum | LTα 322 | 0.001 | 0.051 | 1 | 0.822 |
| | | LTα 347 | -0.002 | 2.039 | 1 | 0.153 |
| | | LTα 371 | 0.000 | 0.000 | 1 | 1.000 |
| | | LTα 389 | 0.000 | 0.000 | 1 | 1.000 |
| | | LTα 411 | 0.001 | 0.334 | 1 | 0.563 |
| | | LTα 488 | 0.009 | 0.287 | 1 | 0.592 |
| | | LTα 525 | 0.046 | 12.480 | 2 | **0.00195** |
| | | site | 0.124 | 30.689 | 2 | <0.001 |
| | | host body mass | 0.023 | 2.779 | 1 | 0.095 |

located near 3' end of the third exon. In IFNβ1, positively selected codon 39 comprised SNPs *IFNβ1* 127 that affected intensity of infection with the nematode *H. glareoli*. Two codons in *TNF* and one in *IFNβ1* displayed signatures of negative selection. The negatively selected codon 31 in *IFNβ1* comprised variant *IFNβ1* 105, which was associated with prevalence and abundance of *H. glareoli*.

# 4. Discussion

## 4.1. Associations between cytokine variants and parasite load

In the present study we found associations between SNPs in cytokines *LTα and IFNβ1* and susceptibility to helminth infections but not to microparasites (bacteria and protists). Intronic variant LTα 535 affected abundance of the nematode *Heligmosomum mixtum*. IFNβ1 105 affected prevanlence and abundance of *H. glareoli*, and IFNβ1 127 affected risk of infection with this parasite.

Despite numerous examples showing links between SNP variation in cytokines with a variety of diseases and conditions (reviewed in [30, 52]), the role of these cell-signalling proteins in resistance against infections in free-living hosts has rarely been studied. In the present paper, we show that variation in the genes coding for two cytokines–lymphotoxin *LTα* and interferon beta *IFNβ1* –affect susceptibility to nematode infections in bank voles. To our knowledge, this is a first report showing they play a role in resistance against infections in free-living species. We hypothesise that the mechanism may be linked to the immunomodulatory effect of nematodes altering production of inflammatory cytokines, yet this hypothesis need further studies.

Among the three frequent bank vole nematodes analysed here, *H. glareoli* lives in close contact with the intestinal wall, likely feeding on mucus and blood, while the larger *H. mixtum*

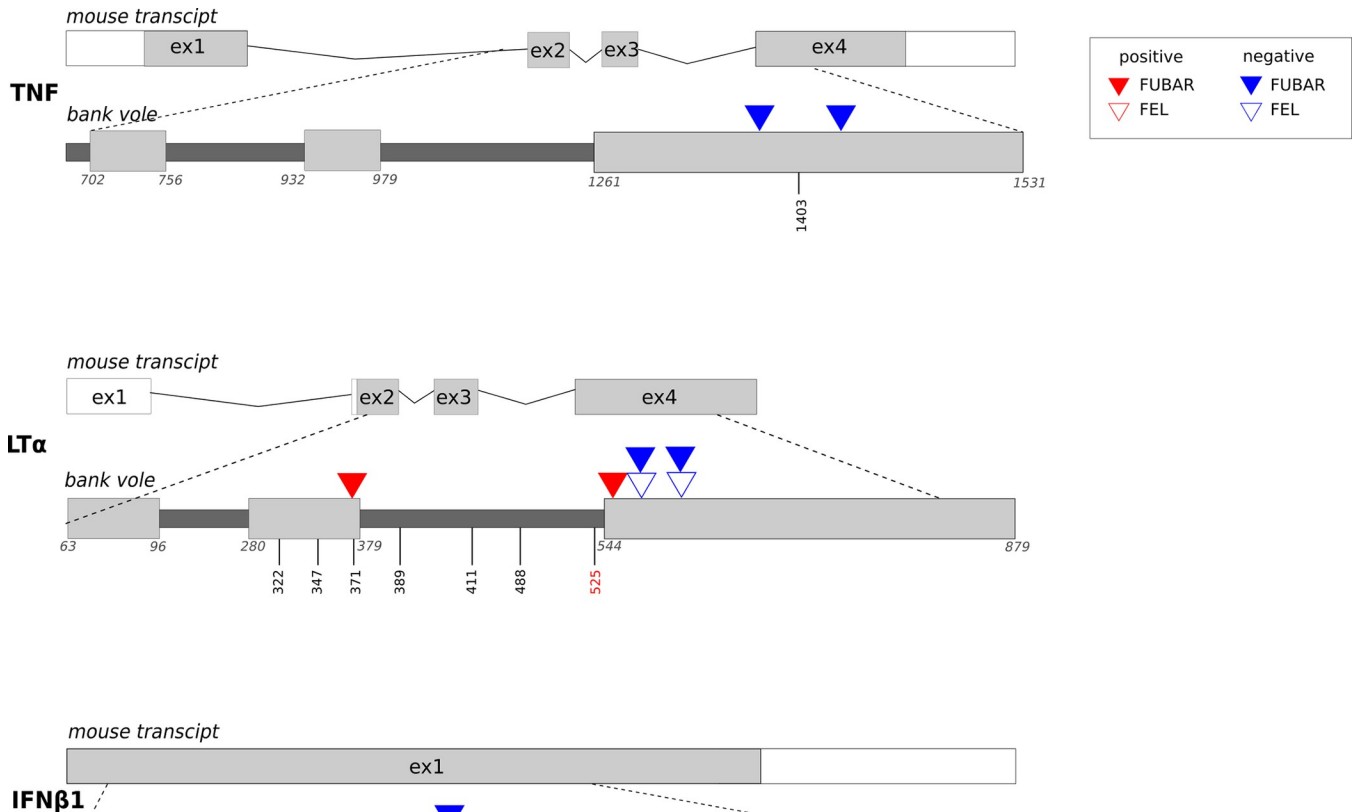

**Fig 2. Schematic position of the sites under selection (marked as triangles) in the studied genes in relation to exon-intron structure.** Positions of SNPs included in the GLM models are marked with bars and numbers, and in red we marked positions of SNPs that were significantly associated with parasite load. In bank vole sequences the numbers show distance in base pairs from the first transcribed nucleotide in mouse CDS. Note that the length of introns differed between mouse and vole.

dwells in the intestine lumen [53]. These two parasite species occur in voles in relatively low numbers, rarely exceeding 10–15 worms. In contrast, infections with *A. tianjinesis* occupying the caecum are often abundant, reaching dozens or hundreds of worms. Such a high parasite burden may result in pathological changes in the guts [54] and lead to depauperated gut microbiota diversity [25], which can have a negative impact on host fitness. In our previous work with the same study system, we found associations between infections with *A. tianjinesis* and genotype in MHC-DRB [21]. This effect was site-dependent: the same variant had the opposite effect at different sites. In our previous studies, we focused on haplotypes rather than SNP variants.

The signatures of positive selection and lower parasite load in *LTα* homozygotes reported in the current study suggest the overdominance of one allele which forms a putatively beneficial genotype. Due to the versatile role of *LTα* in the immune response, it is difficult to provide a functional explanation for the observed polymorphisms. Yet, *LTα* is crucial for development of Peyer's patches–groupings of lymphoid follicles in the mucus membrane lining the small intestine. In mice infected with the nematode *Heligmosomum polygyrus*, *LTα* signalling was essential for the generation of T cells triggering interleukin Il-4 production [55]. *IFNβ*, on the

other hand, is generally linked to antiviral resistance but its activation is mediated by *LTα* [14], which may explain its links to resistance against gastro-intestinal infections reported in the current paper. Contrary to expectations, we did not find an effect of variation within the TNF gene on susceptibility to infections with neither nematodes nor microparasites. TNF triggers an inflammatory, Th-1 type immune response that is usually suppressed by nematode infections [32], which may explain our results.

It is important to note that association studies of wild systems, as the one presented here, have some caveats. First of all, the multiple comparisons may produce false positives. As wild systems are far from randomized trials, it is difficult to control underlying population structure or standing genetic variation, even though we incorporated factors such as sampling site to our association models. To strengthen our results, in S8 Table we present models with all non-genetic terms fitted (rather than only those that significantly affect parasite load), and it is clear that these do not affect the links between genetic variants and susceptibility to parasitic infection. Nonetheless, the candidate genes approach presented here is just a first step to show the role of genetic variation within immunity genes for pathogen resistance but to better understand this connection, further studies should functionally verify the effect of predicted SNPs.

## 4.2. Signatures of selection

Most studies of selection acting on immunity genes in wild mammals have focused on proteins presenting motifs derived from pathogens, such as MHC or TLR, where nucleotide composition was directly attributed to functional variation. In cytokines, identifying potential targets for selection is even more difficult. Some authors have suggested that molecules of such a function are primarily affected by purifying selection [11], but several studies in free-living mammals have reported otherwise [12, 16].

We found signatures of positive selection in two variants: *LTα* 371 and *IFNβ1* 127, which suggests evolutionary pressure favouring these sites. On the other hand, the variant *IFNβ1* 105 displayed signatures of purifying selection, and individuals with the genotype TT were more susceptible to the nematode *H. glareoli*. This may be explained by a significant effect of these variants on parasite load, which is consistent with the hypothesis of pathogen-mediated model of evolution through frequency-dependent selection. It is important to note that signatures of selection do not always imply that the selected site is of functional importance, and some variation on the sequence level may not be ultimately adaptive [56]. The inference about functional importance of the sites under selection may be strengthened by analysing their associations with parasite susceptibility or resistance [57]. Detecting such a link suggests contemporary parasite-driven selection operating at those loci. In our study, both codons displaying signatures of selection in *IFNβ1* comprised SNPs that were significantly associated with parasite load what allows for contributing the signatures of selection to the evolutionary pressure from parasites. However, we are aware that this is a correlative evidence which may arose from an effect of SNPs located in a physically linked region. To strengthen our hypothesis, candidate SNPs should be verified by functional in vitro testing.

## 4.3. Role of non-coding and synonymous variance

Variant *IFNβ1* 105, affecting parasite load, is synonymous. This result may seem confusing, as synonymous substitutions do not affect the amino-acid composition of a protein. However, site-specific signals of selection in this locus strongly suggest that non-neutral pressure is exerted on these sites. The role of non-coding variants may be more significant than previously thought; for instance, the list of human diseases associated with synonymous mutations is expanding [58]. In a large meta-analysis of human GWAS, [59] reported that synonymous

SNPs were as often involved in disease mechanisms as non-synonymous SNPs, and they were not in linkage disequilibrium with causal non-synonymous SNPs. Several mechanisms may explain the role of synonymous mutations. They may affect mRNA splicing and the stability of transcripts [60]. Although coding for the same amino-acid, some variants may be preferred during elongation, promoting co-evolution to optimise translation efficiency [61]. On the other hand, since synonymous mutations are assumed to be evolutionarily silent, their effects might have been under-reported [58, 59]. This further underlines the importance of studies on synonymous SNPs in non-model species for a better understanding of processes maintaining genetic diversity within the immune system in the wild.

Another type of non-coding SNPs that we found to significantly affect the parasite load was an intronic variant *LTα* 525. Notably, it was not linked to any exonic variant, and strong LD was found only to another intronic SNP. Again, human association studies confirmed the role of intronic polymorphisms in susceptibility to diseases [62, 63], particularly when mutations are located close to intron-exon junctions or within a branchpoint sequence [64]. Intronic variants may also affect expression through alternative splicing or interactions with regulatory elements [63]. The intronic *LTα* variant 525 that affected susceptibility to infection with *H. mixtum* was located only 18bp from the intron-exon junction. Introns in the *LTα* gene have been shown to affect expression in several in vitro and in vivo studies (reviewed in [65]), and intronic SNPs can still affect splicing or expression, even if separated by over 30bp from any splice site [63, 66].

Studies of the effect of intronic variants on resistance against infections in the wild are rare, but in humans, positively selected SNP associated with susceptibility to Lassa virus have been found in the interleukin *IL21* gene outside the open-reading frame [67]. The authors suggested that those variants may lead to regulatory changes such as differential gene expression. Intronic variants may also affect cytokine interactions with other components of the immune system. For instance, an intronic variant in the human *IFNγ* gene coincides with a putative NF-κB binding site which might have functional consequences for the transcription of the human *IFNγ* gene [68]. In bank voles, SNP located within the promoter of the TNF gene affected susceptibility to Puumala virus (PUUV) [17]. Unfortunately, we could not confirm this pattern, as the fragment amplified in the current study did not span the 5' upstream region.

## 5. Conclusion

In the present paper we examined parasite-driven selection in cytokines, excreted molecules involved in signal transduction. We identified SNPs affecting parasite load with intestinal nematodes, and we showed that codons comprising those SNPs display signatures of selection. Importantly, among those variants we found one located in an intron, and another one coding for synonymous substitution. Such variants had been commonly disregarded in studies of the genetic background of host-parasite co-evolution, yet our results show that they play a role in parasite resistance in wild systems. We propose that non-coding variants should not be automatically considered non-functional, and only by including them in association studies are we able to understand the genetic background of parasite resistance in the wild.

## Supporting information

**S1 Table. PCR conditions and sequences of the primers used in the current study.** The primers included degenerated sites (marked in bold).
(PDF)

**S2 Table. Characteristic of the studied amplicons and summary of polymorphisms within the studied genes.** Number of respective exons and introns in mouse is given in parentheses.

In *TNF* and *LTα* after slash we provide polymorphism summarises for exonic parts only.
(PDF)

**S3 Table. Number of voles with given genotypes after filtering.** We removed missing calls, variants with MAF<0.05, not in Hardy-Weinberg equilibrium (threshold p<0.001), and in linkage disequilibrium (r> 0.7).
(PDF)

**S4 Table. Prevalence of infections among bank voles.** The number of infected animals differs between studied genes because not all individuals were genotyped in three loci. non-inf–number of non-infected hosts, inf–number of infected hosts, %–percentage of host infected.
(PDF)

**S5 Table. Summary of effect of non-genetic terms on parasite load.** Terms with p<0.1 (marked in bold) were included in GLM models testing for the effect of genetic variance (S7 Table). The models were run separately on three datasets, each including voles genotyped at a given locus (not all animals were genotyped in all loci).
(PDF)

**S6 Table. Effect on cytokine genetic variants on parasite load.** As response variables we used only pathogens that infected 20–80% of hosts (S4 Table). β is parameter estimate for each contrast, $R^2$ is partial coefficient of determination (effect size), $\chi^2$ and p-values are based on LR type III test. To control for multiple comparisons when testing for the effect of several genetic variants, we used conservative Bonferroni correction; for 10 genetic terms (SNPs) tested, the critical p-level corresponding to α = 0.05 was 0.005. Exact p-values of genetic terms significant after correction are given in bold.
(PDF)

**S7 Table. Codons under selection.** Only exonic parts of the studied genes are analysed. Codons were numbered starting from the first genotyped nucleotide, not from the first transcribed nucleotide. Names of the corresponding SNPs, as used in the current paper, are given in brackets. Codons that comprised SNPs significantly associated with the parasite load are given in bold.
(PDF)

**S8 Table. Effect on cytokine genetic variants on parasite load with all non-genetic terms fitted.** β is parameter estimate for each contrast, $R^2$ is partial coefficient of determination (effect size), $\chi^2$ and p-values are based on LR type III test. To control for multiple comparisons when testing for the effect of several genetic variants, we used conservative Bonferroni correction; for 10 genetic terms (SNPs) tested, the critical p-level corresponding to α = 0.05 was 0.005. Exact p-values of genetic terms significant after correction are given in bold.
(PDF)

**S1 Fig.** Parasite load by parasite species in individuals genotyped in a) *TNF*, b) *LTα*, and c) *IFNβ1*. Dark bars represent infected animals, light–non-infected.
(PDF)

## Acknowledgments

We thank D.R. Laetsch and M.A. Wenzel for their valuable hints on the bioinformatic pipeline and data analysis. We are thankful to W. Babik who provided access to an Illumina MiSeq platform, and to K. Dudek who prepared the Nextera library.

## Author Contributions

**Conceptualization:** Agnieszka Kloch.

**Formal analysis:** Agnieszka Kloch, Renata Welc-Falęciak, Anna Bajer.

**Funding acquisition:** Agnieszka Kloch.

**Investigation:** Agnieszka Kloch, Ewa J. Mierzejewska, Renata Welc-Falęciak, Anna Bajer, Aleksandra Biedrzycka.

**Methodology:** Agnieszka Kloch, Anna Bajer, Aleksandra Biedrzycka.

**Project administration:** Agnieszka Kloch.

**Visualization:** Agnieszka Kloch.

**Writing – original draft:** Agnieszka Kloch, Aleksandra Biedrzycka.

**Writing – review & editing:** Agnieszka Kloch.

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
