## [Decision Letter · Decision Letter 0]

10 May 2022

PONE-D-21-29710Cytokine gene polymorphism and parasite susceptibility in free-living rodents: importance of non-coding variantsPLOS ONE

Dear Dr. Kloch,

Thank you for submitting your manuscript to PLOS ONE. After careful consideration, we feel that it has merit but does not fully meet PLOS ONE’s publication criteria as it currently stands. Therefore, we invite you to submit a revised version of the manuscript that addresses the points raised during the review process. More precisely, the three reviewers agree that your manuscript is interesting but need of major revision before a possible acceptance.

A first comment concerns the sampling cohort, which is quite small and which is heterogeneous (recruitment from 2005 to 2016). It is not therefore easy to compare data from different years and locations. Different other factor could also play a role to explain the obtained results. To correct these risks, I agree with the first reviewer and it would be important to strengthen multiple test correction. And accordingly, to rewrite abstract, results and discussion.

Moreover, I also agree with the second and third reviewers who propose to strengthen the introduction, giving more information concerning the study system, the interest to associate these particular cytokines genes with host-parasite interactions and also to better explain your aims and hypothesis in the last paragraph of the introduction and discuss mechanisms of balancing selection.

Like for the three reviewers, I also think that it would be good to clarify the method chapter and particularly the statistical analyses.

All reviewers noted several grammatical and typos errors. They ask to rewrite the text with a native English speaker. I agree with their comments and I think that the text would be improved too.

Finally, all minor comments noted by the three reviewers will have to be taken into account. This will really improve the quality of the text.

We look forward to receiving your revised manuscript.

Kind regards,

Johan R. Michaux

Academic Editor

PLOS ONE

Journal Requirements:

2. In your Methods section, please provide additional location information, including geographic coordinates of your field collection site if available.

4. To comply with PLOS ONE submissions requirements, please provide methods of sacrifice in the Methods section of your manuscript.

5. Thank you for stating the following financial disclosure: "The work was supported by grant no. 2012/07/B/NZ8/00058 from the Polish National Science Centre to AK."

7. We note that you have stated that you will provide repository information for your data at acceptance. Should your manuscript be accepted for publication, we will hold it until you provide the relevant accession numbers or DOIs necessary to access your data. If you wish to make changes to your Data Availability statement, please describe these changes in your cover letter and we will update your Data Availability statement to reflect the information you provide.

Reviewers' comments:

Reviewer's Responses to Questions

**Comments to the Author**

1. Is the manuscript technically sound, and do the data support the conclusions?

Reviewer #1: Yes

Reviewer #2: No

Reviewer #3: Yes

2. Has the statistical analysis been performed appropriately and rigorously? 

Reviewer #1: Yes

Reviewer #2: No

Reviewer #3: Yes

3. Have the authors made all data underlying the findings in their manuscript fully available?

Reviewer #1: Yes

Reviewer #2: Yes

Reviewer #3: No

4. Is the manuscript presented in an intelligible fashion and written in standard English?

Reviewer #1: No

Reviewer #2: Yes

Reviewer #3: Yes

5. Review Comments to the Author

Reviewer #1: General comments

This is an interesting study focused on the variation of three cytokines in bank voles. I particularly acknowledge that authors try to correlate particular SNP variation with infection/ diseases data, which is very rare outside human studies. I also agree with the authors that further research should focus more on genes outside MHC and TLRs and/or non-coding regions.

My major comments are that authors should better explain their aims and hypothesis in the last paragraph of the introduction and discuss mechanisms of balancing selection. To avoid multiple testing issues, I am not sure whether it would be better to merge models from Table S6 with models in Table 7 rather than including only significant and marginally significant predictors from models in Table S6 to the final models in Table S7. However, I am leaving the decision to the editor. I also think that manuscript should be checked by an English linguist.

Please, include the line numbering next time as it helps to orient while reviewing the manuscript. I had to mostly calculate line numbers by hand as the manuscript is in PDF format and also copy the whole sentences that need to be modified…

Specific line-by-line comments

***abstract

*P1, L9-L11 Please, revise the sentence below – some words are missing in ““ part

Two SNPs in LTα and two in IFNβ1 significantly affected susceptibility to nematodes, “and was of them was also associated with susceptibility“ to microbial pathogen Bartonella.

*P1, L3-L4

MHC is not a receptor. Revise the sentence below for accuracy:

Despite extensive studies of receptors, such as MHC or TLR, little is known about efferent arm of the immune system.

***Introduction

*P3, L3 What are higher vertebrates? Maybe mentioning mammals is redundant in: In mammals and higher vertebrates, the immune system comprises dozens of interacting molecules, yet the mechanisms of this selection have been comprehensively studied only in the case of major histocompatibility complex (MHC) genes (e.g. Radwan et al 2020).

*P3, L11

Typing error: e.g.

in eg. Fornuskova et al. 2013, Babik et al. 2015, Kloch et al. 2018)

*P3, L16-18 Maybe abbreviation mentioned for the first time should be explained already here:

For instance, balancing selection was found in interleukins Il-1B, Il-2, and TNF in field voles (Turner et al. 2012), and in Il-10 and CD14 in humans (Ferrer-Atmetlla et al. 2008).

and further in L20, In humans, polymorphism within the LTα has been linked to several diseases, such as Mycobacterium leprae (Ware 2005) and malaria (Barbier et al. 2008)

*L27, Please, revise the first part of the sentence for clarity. Gene name abbreviations are not necessary to be explained here. Also, all gene names should be written in Italics here and in other places in the manuscript where they mean gene products and not proteins.

To better understand the role of the parasite-driven selection maintaining polymorphism in cytokines, and to further explain the role of this polymorphism in resistance against pathogens in the wild, we studied three cytokine genes: tumor necrosis factor (TNF), lymphotoxin alpha (LTα) formerly known as tumor necrosis factor beta (TNF�), and interferon beta (IFNβ1).

*I think that the aims of the study and hypothesis should be better clarified in the last paragraph of the introduction.

*I would also briefly mention the mechanism of parasite-mediated balancing selection in the introduction: heterozygote advantage, NFDS, and space-time fluctuating selection.

***Methods

*P5, L23, What reference sequences were used for numbering? Please, mention their sequence IDs.

*P5, L25: The following sentence should be better in results.

The sequencing revealed 43 SNP in ~2000bp in total

*P4, L26

I am lacking details about which tissue has been used for DNA extraction and which extraction kit has been used. It should be here even if it is referenced in one of the author´s papers cited.

*P5, L24

I am not sure if I fully understand the sequencing strategy. Do authors have only one amplicon per gene? But given the restricted overlaps between forward and reverse reads/ or little variation in the overlapping regions, there was a need to phase the alleles? If so, please mention it explicitly in the methods. What is the percentage of alleles that were directly reconstructed without the need of phasing?

*P6, L1

What is MAF? Please, explain. Please, also better mention what is the variant frequency in % than less in than 5 animals.

*P6, L3

I appreciate that the authors used Benjamini-Yekutieli false discovery rate to avoid type I error. However, to avoid multiple testing issues, I am not sure whether it would be better to merge models from Table S6 with models in Table 7 rather than including only significant and marginally significant predictors from models in Table S6 to the final models in Table S7.

*P6, L20

I do not see any results for recombination testing and given that I think that recombination was not revealed. Please, mention it either here or in the relevant result section.

*P6, L21

Please, add that the selection methods used are based on dN/dS ratio testing.

***Results

*P6, 3.1. Cytokine polymorphisms and susceptibility to infections

I would add some general summary of polymorphism revealed, ideally with a table in SI with values such as number of sequences, number of unique nucleotide alleles, number of variable sites, number of substitutions to see how much are cytokines variable.

*** Discussion

*P7, L30

MHC is not a receptor. Please, correct the following sentence:

Studies of associations between polymorphisms in the immunity genes and susceptibility to diseases in wild mammals usually focus on receptor proteins, such as MHC or TLR

*P7, L30 I do not agree with the statement that nucleotide variation in molecules that physically interact with pathogen structure is ultimately functional. There is much variation that is the most likely non-adaptive, even if it is under positive selection. Please, see e.g. Těšický et al. 2020 wherein some approaches on how to distinguish putative functional variation from non-functional one are outlined.

Těšický M, Velová H, Novotný M, et al (2020) Positive selection and convergent evolution shape molecular phenotypic traits of innate immunity receptors in tits (Paridae). Mol Ecol 3056–3070. https://doi.org/10.1111/mec.15547

Original sentence: Since they physically interact with pathogen-derived motifs, their nucleotide composition can be directly attributed to functional variation

*P8, L7 Please, clarify what is meant by repeated rounds of positive selection interspersed with purifying selection.

***Tables and figures

*Figure 1

I think that it is quite confusing to present in upper panels of the figure both percentage and no. individuals as there is no scale on the y-axis for the percentage. Also, e.g. column for 57 % is higher than for 34.4 %. Why there is no CC genotype for LTalfa 322?

*Figure 2.

It seems that the last sentence of figure 1 caption is unfinished: “Positions of SNPs are given with bars, and SNPs”

Please, also add the numbering at the beginnings and the ends of exons.

* Table S6 and S7

Please add a new column with the number of observations in both tables

*Table S8

LTalfa 107* - what does this asterisk mean?

Please, include the same position identified under selection by multiple methods in the same row to allow a reader to better compare which positions have been identified by the multiple selection methods.

Reviewer #2: In the paper " Cytokine gene polymorphism and parasite susceptibility in free-living rodents: importance of non-coding variants”, Agnieszka Kloch and co-Authors investigated how genetic variation in cytokines affects susceptibility to parasitic diseases in bank voles.

The Authors, in particular, studied the three cytokines TNF, LTα and IFNβ1, demonstrating that two SNPs in LTα and two in IFNβ1 significantly affected susceptibility to nematodes and to the microbial pathogen Bartonella.

The Authors concluded that the identified cytokines are prone to parasite-driven selection, and non-coding variants may be linked to susceptibility to infections in wild systems.

The cohort is quite small for this particular type of study; then, the data are quite heterogeneous, because the recruitment was performed in 2005 and 2016; it is not easy to compare data from different years and locations.

The Authors wrote they corrected the results by means of false discovery rate; nevertheless, in the present paper there are a lot of comparisons and given the great number of tests performed in the study, some obtained p-values may be spurious. I therefore ask the Authors to strengthen multiple test correction; accordingly, the Authors should rewrite abstract, results and discussion. I recognise that if the Authors would apply a too strong correction (such as Bonferroni), probably most of the comparisons would be lost; so, I ask the Authors to set a value in order to clean the P which may be resulted significant only by chance. In fact, in my opinion, P-values of 0.02, 0.03 obtained in a small cohort study without a strong correction for multiple testing may be spurious.

Minor changes:

• Introduction, L3: please change the word “dozen” with another more scientific word

• Introduction: in the sentence “In humans, polymorphism within the LTα …”, please define LT the first time you mention it.

• Material and Methods: I ask the Authors a brief comment in the main-text about the sentence: “samples from 2005 were sequenced using 150 cycles, and samples from 2016 using 300 cycles.

• Material and Methods: In the sentence: “In a second step, we excluded from the model variants with MAF<0.5 and those present in fewer than 5 animals (see Table S4), as we lacked statistical power to test for their effects”, I understand the reasons of a lack of statistical power, but, in my opinion, a new mutation may be (at the beginning) present in only one animal and with a MAF<0.5. Please comment this fact in the main-text.

• Supplementary Tables 2: the Authors indicated that some primers contain degenerate nucleotides (such as W, Y, etc.); I ask the Authors to add a brief comment and an explanation of this in the main-text

In conclusion, in my opinion the present paper cannot be accepted in the present form; I ask the Authors both major and minor revisions. A major rewrite is necessary of all the sections of the paper.

Reviewer #3: In this manuscript, the authors investigate associations between genetic variants in cytokine genes and parasite infection and parasite load in wild bank voles. The authors find that there are associations between a few SNPs and parasite infection or load and there is evidence that these variants are under selection.

I believe that the topic is interesting, particularly because studies looking beyond the MHC and in wild systems are rare. However, I do feel that the introduction could be strengthened by giving a bit more background to the study system and stating the rationale for looking at these particular cytokine genes in this host-parasite system. In addition, I think the methods could be a bit clearer on the details of the study system and statistical analyses. There are also some grammatical errors and typos, which I have tried to point out as best as I can despite the lack of line numbers. I go through these points in more detail, along with other comments that I hope will improve the clarity of the manuscript, below:

Abstract, line 3: little is known about ‘the’ efferent arm of the immune system

Abstract: would be nice to add in the sample size into the abstract

Abstract, line 9: I think this should read ‘one of them’ rather than ‘was of them’

Abstract, line 10: would be nice to add in the type of selection observed

Abstract, final sentence: it appears that there are more associations with coding (exonic) variants that affect parasite infection/load than intronic variants?

Introduction, paragraph 2, line 3: I think the ‘and’ needs to be removed here before ‘they are expected’ and ‘have’ needs to be added before ‘reported signatures’

Introduction, paragraph 2, line 6 and 11: might need to specify that selection was found in the genes (line 6) and variation was genetic variants in interleukin genes (line 11)

Introduction, paragraph 2, line 12: find = found

Introduction, paragraph 2, line 12: ‘A’ single study. Might need to specify whether this has been investigated and not found, or if only one study has investigated non-coding regions

Introduction, paragraph 2: in the next paragraph you explain what LTα is but it would be better to put this here on the first mention of this cytokine gene.

Introduction, paragraph 3, first sentence: remove ‘the’ before parasite-driven selection and ‘this polymorphism’ should be ‘these polymorphisms’

Introduction, paragraph 3, second sentence: remove ‘an’ before evidence, its importance = their importance and ‘makes them a promising candidate gene’ should be ‘makes them promising candidate genes’

Introduction, paragraph 3, third sentence: add ‘the’ before acute phase and remove ‘the’ before inflammation

Introduction: I believe that the introduction would benefit from bringing in the study system and describing the types of pathogens that are present in the two populations of bank voles. What pathogens cause more morbidity or mortality in this system? It would then be nice to use this information to link into why you are studying these particular cytokine genes so that it feels more hypothesis driven. This would also help you to develop key expectations of the types of genes that might be important – from previous work for example by Turner et al in the field voles – and also what types of variants might be important (e.g. synonymous vs non-synonymous and exonic vs intronic).

Materials and methods, section 2.1: It would be important to add here the sample size of voles in the two years at each site into the main text (rather than a supplementary table). Later, post QC of SNPs it would be good to state from X number of individuals X were used in the final analyses.

Materials and methods, section 2.1: Are any of the individuals in these populations related? Did you control for this at all?

Materials and methods, section 2.1: A brief discussion of the differences in parasite burden between the two sites would be good to have here (or in the introduction) so that this manuscript can stand alone.

Materials and methods, section 2.1: In the introduction and discussion it would be important to introduce and compare respectively the associations you find here with those in the MHC and TLR genes in the same study system – are patterns of synonymous vs non-synonymous and exonic vs intronic similar? Are effect sizes larger in the MHC or TLR genes than in the cytokine genes?

Materials and methods, section 2.1, second to last sentence: ‘the’ protocol

Materials and methods, page 5, third sentence: repeat of the word ‘using’

Materials and methods, page 5, first paragraph: why were a different number of cycles used?

Materials and methods, page 5, second paragraph, second sentence: ‘a’ reference and ‘genes of interest’

Materials and methods, page 5, second paragraph: it would help to give a version and reference for both the bank vole genome and mouse genome used (for orthologues).

Materials and methods, page 5, second paragraph: it would help the reader if you explained what the abbreviations are that are used as your QC criteria in vcffilter.

Materials and methods, page 5, line 12: ‘where physical position’ rather than ‘were physical position’

Materials and methods, page 5, second paragraph, final sentence: would be nice to add in the number of genes and corresponding chromosomes

Materials and methods, page 5, section 2.3, first paragraph: I don’t really understand the last sentence about minimising false positives– could you clarify?

Materials and methods, page 5, section 2.3, second paragraph: do you mean you removed SNPs with a genotyping rate <100% and any individuals that had any missing SNP genotypes? Please give final sample sizes after QC

Materials and methods, page 5, section 2.3, second paragraph: How many SNPs deviated from HWE? This might be expected if selection is occurring on these loci.

Materials and methods, page 6, first paragraph: I think there is a typo – surely minor allele frequency must be <0.5?

Materials and methods, page 6, first paragraph: rather than put the pathogens in a supplementary table it would be important to specifically list each pathogen looked at and ideally the % prevalence. Histograms of parasite load would be nice and should go in supplement. I think this is really important because it is not immediately obvious to the reader how many pathogens you are looking at and therefore it is hard to put the results into context.

Materials and methods, page 6, second paragraph: some details of the models are missing including the type of link, how significance of fixed effects was determined, how post-hoc testing was done between SNP genotypes, which package was used for models etc. How did you control for overdispersion (and presumably zero-inflation) in the poisson model? Or did the model fit without other means of controlling for overdispersion? Did every pathogen abundance/load model suitably fit a poisson model? Did you try running these in a hurdle model which should combine the two models you mention (parasite infection y/n and parasite load)? Did you test for any interactions between SNPs and non-genetic sources such as sex or site?

Materials and methods, page 6, section 2.4: it would be useful to the reader if you could explain how these codon-based tests identify sites under selection, even briefly.

Results: I think it would be useful to explicitly state each of the null results (for both models and each pathogen) you found too rather than just the significant results. It would also be good to state X number of SNPs out of X total SNPs in a gene were associated with a trait – as you did with TNF in the first sentence. It would also be good to state whether SNP effects were additive, and to compare the effects of different SNPs (on the same parasite) in the discussion.

Results: There is some inconsistency in how you explain the results. For some SNPs you give the average number of worms in each genotype class but not for others – it would be good to provide this for all. Also it would be good to have errors around these values and to explain how these were calculated in the methods – is it from the raw data or post hoc testing controlling for other non-genetic variables in the model?

Results: I would not discuss a result that did not meet the p value after correction for multiple testing.

Results: It would be nice to have full results from each model (model estimates, errors and p values for each SNP and for non-genetic variables too and post hoc testing between genotypes) reported somewhere, even if in the supplement. You could also compare the effects of non-genetic sources to SNPs in each model

Results, page 6, second paragraph: give full latin name on first mention

Results, page 7, second paragraph: typo – lowerst

Results, page 7, second paragraph: what does average high mean?

Results, page 7, section 3.2: typo compring = comprising

Figure 1: Explain bar chart colours and outliers and labels. A stacked bar chart might be a better option.

Figure 2: I think the caption is unfinished?

Discussion: it would be nice if the discussion was reformatted to follow the same format as the results – starting with SNPs associated with infection or parasite load, followed by discussion of selection on these variants

Discussion, page 8, line 3: I would rephrase to ‘genetic variation in cytokine genes also plays’ rather than ‘cytokine variance’ as this sounds like you are actually measuring cytokine levels

Discussion, page 8, paragraph 2: there are a lot of ‘what’s in this paragraph that should be ‘which’

Discussion, page 8, paragraph 2: I wouldn’t comment on the effect on Bartonella sp. as I think it was non significant after correction for multiple testing

Discussion, page 8, paragraph 2: following on from the final sentence it would be nice to have some insight into how these pathogens affect morbidity and mortality in this species or a similar species

Discussion, page 8, paragraph 3: This is a nice discussion of the role of synonymous mutations

Discussion: some more discussion of other wild systems (e.g. Turner et al’s work in the field voles) and comparisons to effects of the MHC and TLR variants previously investigated in this system would strengthen the manuscript. It would also be nice to add whether variants in IFNβ1 and LTα (or orthologues) have previously been implicated in susceptibility to parasitic diseases in other species.

Discussion, page 9: the conclusion feels a bit rushed. While there is an intronic and a synonymous SNP, it appears that more associations with exonic and non-synonymous SNPs and parasite infection/load – so can you say that intronic and synonymous variants play a (more) important role? Or are you just saying that they are also present and should not be ruled out?

Discussion: some discussion of the advantages and disadvantages of candidate gene studies would be helpful to the reader

6. PLOS authors have the option to publish the peer review history of their article (what does this mean?). If published, this will include your full peer review and any attached files.

Reviewer #1: No

Reviewer #2: No

Reviewer #3: No

---

## [Author Response · Author response to Decision Letter 0]

7 Jul 2022

Response to the Reviewer #1:

My major comments are that authors should better explain their aims and hypothesis in the last paragraph of the introduction and discuss mechanisms of balancing selection. 

We expanded the Introduction adding a passage describing the study system and we specified the aims and the hypothesis. We also completed the Introduction with a passage discussing mechanisms of the BS.

To avoid multiple testing issues, I am not sure whether it would be better to merge models from Table S6 with models in Table 7 rather than including only significant and marginally significant predictors from models in Table S6 to the final models in Table S7. 

The two step procedure, where first non-genetic minimal model is constructed and then genetic terms are fitted, has been often used before (eg. Paterson et al. PNAS 1998: 95, Turner et al. PloS Genetics 2011). It helps to avoid overfitting the model and to improve the fit of the model by including only those non-genetic factors that indeed affect parasite load. We tested two alternative procedures: i) model simplification based on AIC and removing variables that explain little variance in data, and ii) fitting non-genetic terms as fixed variables in mixed models, and both gave similar results as the original models but resulting models had lower fit. 

Specific line-by-line comments

***abstract

*P1, L9-L11 Please, revise the sentence below – some words are missing in ““ part

Two SNPs in LTα and two in IFNβ1 significantly affected susceptibility to nematodes, “and was of them was also associated with susceptibility“ to microbial pathogen Bartonella.

 The sentence was rewritten, and the part about Bartonella was removed as after applying Bonferroni correction this association turned non-significant.

*P1, L3-L4

MHC is not a receptor. Revise the sentence below for accuracy:

Despite extensive studies of receptors, such as MHC or TLR, little is known about efferent arm of the immune system.

 We changed “receptor” to a more general term “proteins interacting with pathogen-derived ligands”

***Introduction

*P3, L3 What are higher vertebrates? Maybe mentioning mammals is redundant in: In mammals and higher vertebrates, the immune system comprises dozens of interacting molecules, yet the mechanisms of this selection have been comprehensively studied only in the case of major histocompatibility complex (MHC) genes (e.g. Radwan et al 2020).

 Since MHC has been also studied in fish, we decided this sentence may be misleading and we changed “ In mammals and higher vertebrates” simply to “in vertebrates”.

*P3, L16-18 Maybe abbreviation mentioned for the first time should be explained already here:

For instance, balancing selection was found in interleukins Il-1B, Il-2, and TNF in field voles (Turner et al. 2012), and in Il-10 and CD14 in humans (Ferrer-Atmetlla et al. 2008).

and further in L20, In humans, polymorphism within the LTα has been linked to several diseases, such as Mycobacterium leprae (Ware 2005) and malaria (Barbier et al. 2008)

 We added the full names of these proteins.

*L27, Please, revise the first part of the sentence for clarity. Gene name abbreviations are not necessary to be explained here. Also, all gene names should be written in Italics here and in other places in the manuscript where they mean gene products and not proteins.

To better understand the role of the parasite-driven selection maintaining polymorphism in cytokines, and to further explain the role of this polymorphism in resistance against pathogens in the wild, we studied three cytokine genes: tumor necrosis factor (TNF), lymphotoxin alpha (LTα) formerly known as tumor necrosis factor beta (TNFβ), and interferon beta (IFNβ1).

After re-arranging the Introduction, we provided full names of the genes in their first occurrence in the text (except for abstract). We changed abbreviations to italics throughout the paper to make clear that we refer to a gene, and not its product.

*I think that the aims of the study and hypothesis should be better clarified in the last paragraph of the introduction.

We clarified the hypothesis and aims

*I would also briefly mention the mechanism of parasite-mediated balancing selection in the introduction: heterozygote advantage, NFDS, and space-time fluctuating selection.

 We added a relevant passage to the first paragraph of the Introduction

***Methods

*P5, L23, What reference sequences were used for numbering? Please, mention their sequence IDs.

 We added GenBank accesion numbers for the reference sequences. 

*P5, L25: The following sentence should be better in results.

The sequencing revealed 43 SNP in ~2000bp in total

 We decided to remove this sentence, as we describe the number of SNPs in each gene in details in the section 2.3, paragraph 2.

*P4, L26

I am lacking details about which tissue has been used for DNA extraction and which extraction kit has been used. It should be here even if it is referenced in one of the author´s papers cited.

 We added information on DNA extraction.

*P5, L24

I am not sure if I fully understand the sequencing strategy. Do authors have only one amplicon per gene? But given the restricted overlaps between forward and reverse reads/ or little variation in the overlapping regions, there was a need to phase the alleles? If so, please mention it explicitly in the methods. What is the percentage of alleles that were directly reconstructed without the need of phasing?

In our case, Illumina reads are shorter than the amplicons. The chemistry allows for sequencing fragments up to 300bp, and amplified fragments of LTa and TNF were ~800bp. Thus, prior to sequencing the amplicons had to be split into shorter fragments using Illumina Nextera kit. As a result, some distant SNPs could be always located in separate fragments. Lacking information about the physical position of alleles on a DNA strand, Freebayes (the software used for variant calling and haplotype reconstruction) could not phase some variants. The alleles in those SNPs had to be phased computationally. There were 5 such SNPs out of 36 SNPs in TNF, and 6 of 18 in LTA, and none in IFN. We added this information and we changed potentially confusing phrase “Phased alleles were reconstructed...” to “ Phased alleles were converted to fasta format...”

*P6, L1

What is MAF? Please, explain. Please, also better mention what is the variant frequency in % than less in than 5 animals.

MAF is minor allele frequency, and MAF of 0.05 is a standard filtering threshold. However, in our case due to relatively low sample size after removing alleles with frequency <5%, we still had some variants that were in too few voles to be included in the statistical models. Thus, we applied second filtering step, removing alleles present in fewer that 5 animals to gain enough statistical power. 

*P6, L3

I appreciate that the authors used Benjamini-Yekutieli false discovery rate to avoid type I error. However, to avoid multiple testing issues, I am not sure whether it would be better to merge models from Table S6 with models in Table 7 rather than including only significant and marginally significant predictors from models in Table S6 to the final models in Table S7.

In general, we followed two step procedure used in several association studies of free-living mammals (eg. Paterson et al. PNAS 1998: 95, Turner et al. PloS Genetics 2011) to avoid model overfitting and to obtain better fitting models. We added this explanation to the section 2.3. 

Before submitting the paper, we tested two alternative procedures: i) model simplification based on AIC and removing variables that explain little variance in data, and ii) fitting non-genetic terms as fixed variables in mixed models, and both gave similar results as the original models but resulting models had lower fit. 

*P6, L20

I do not see any results for recombination testing and given that I think that recombination was not revealed. Please, mention it either here or in the relevant result section.

We found no evidence for recombination and we added this sentence in the beginning of section 3.2.

*P6, L21

Please, add that the selection methods used are based on dN/dS ratio testing.

 We added brief description of the tests.

***Results

*P6, 3.1. Cytokine polymorphisms and susceptibility to infections

I would add some general summary of polymorphism revealed, ideally with a table in SI with values such as number of sequences, number of unique nucleotide alleles, number of variable sites, number of substitutions to see how much are cytokines variable.

We rewrite Table (now) S2 adding basing statistics describing nucleotide diversity, such as number of segregating sites (S), number of haplotypes, nucleotide diversity (π) etc, and we summarized the table in the first section of the Results. 

*** Discussion

*P7, L30

MHC is not a receptor. Please, correct the following sentence:

Studies of associations between polymorphisms in the immunity genes and susceptibility to diseases in wild mammals usually focus on receptor proteins, such as MHC or TLR

 We changed “receptor” to a descriptive term : “proteins presenting motifs derived from pathogens”

*P7, L30 I do not agree with the statement that nucleotide variation in molecules that physically interact with pathogen structure is ultimately functional. There is much variation that is the most likely non-adaptive, even if it is under positive selection. Please, see e.g. Těšický et al. 2020 wherein some approaches on how to distinguish putative functional variation from non-functional one are outlined.

Těšický M, Velová H, Novotný M, et al (2020) Positive selection and convergent evolution shape molecular phenotypic traits of innate immunity receptors in tits (Paridae). Mol Ecol 3056–3070. https://doi.org/10.1111/mec.15547

Original sentence: Since they physically interact with pathogen-derived motifs, their nucleotide composition can be directly attributed to functional variation

We thank for this remark and suggesting us the paper. We added this remark to the re-arranged version of the Discussion. 

*P8, L7 Please, clarify what is meant by repeated rounds of positive selection interspersed with purifying selection.

The sentence was a clumsy attempt to describe frequency-dependent selection where a rare allele is supposed to be advantageous and it is favored but once its frequency increases, it is more likely to be “recognized” by pathogens and thus it turns disadvantageous. We altered the sentence so that it better describes FDS.

***Tables and figures

*Figure 1

I think that it is quite confusing to present in upper panels of the figure both percentage and no. individuals as there is no scale on the y-axis for the percentage. Also, e.g. column for 57 % is higher than for 34.4 %. Why there is no CC genotype for LTalfa 322?

We clarified the figure, changing bars to stacked bars and by removing %. The % were meant to show how many individuals with given genotype were infected but we agree that this seemed confusing. 

There was only one individual with the genotype CC in LTa and we removed it from the analysis as we lack statistical power to test the effect of this genotype. We described this in the Results.

*Figure 2.

It seems that the last sentence of figure 1 caption is unfinished: “Positions of SNPs are given with bars, and SNPs”

Please, also add the numbering at the beginnings and the ends of exons.

We completed the sentence (it seems that there was some formatting error and it was not visible) and we added the numbering. 

* Table S6 and S7

Please add a new column with the number of observations in both tables

 We added sample sizes below the gene names. 

*Table S8

LTalfa 107* - what does this asterisk mean?

Please, include the same position identified under selection by multiple methods in the same row to allow a reader to better compare which positions have been identified by the multiple selection methods.

We aligned the positions vertically in the table to facilitate comparisons. We used asterisk to indicated that the positively selected codon 107 is next to codon 108 which corresponds to LT� SNP 322 that was significantly associated with presence of A. tianjinensis. We added this information below the table. 

Response to the Reviewer #2:

The cohort is quite small for this particular type of study; then, the data are quite heterogeneous, because the recruitment was performed in 2005 and 2016; it is not easy to compare data from different years and locations.

We are afraid that this is inherent problem in field studies where obtaining sample sizes comparable to human association studies is difficult and also unethical if many the animals are culled. 

We dealt with the problem the best we could by controlling for data heterogenity in GLMs. In the models, we included all factors that could affect parasite load beside genetic factors eg. year, sample site or host sex.

The Authors wrote they corrected the results by means of false discovery rate; nevertheless, in the present paper there are a lot of comparisons and given the great number of tests performed in the study, some obtained p-values may be spurious. I therefore ask the Authors to strengthen multiple test correction; accordingly, the Authors should rewrite abstract, results and discussion. I recognise that if the Authors would apply a too strong correction (such as Bonferroni), probably most of the comparisons would be lost; so, I ask the Authors to set a value in order to clean the P which may be resulted significant only by chance. In fact, in my opinion, P-values of 0.02, 0.03 obtained in a small cohort study without a strong correction for multiple testing may be spurious.

We applied Bonferroni correction as requested and significant values remained valid except for the association with Bartonella. 

We tested for the effect of 10 SNPs (2 in IFN, 1 in TNF, and 7 in LTa) and in the reviewed manuscript we adapted the most conservative criterion i.e. Bonferroni correction with 10 comparisons, the critical p-level correcponding to α=0.05 is 0.05/10 = 0.005. On the other hand, there is no statistical correction that could be applied when the same set of explanatory variables – alleles in our case – is fitted in different models with different parasite species as response, as this does not meet the definition of multiple comparisons. 

Minor changes:

• Introduction, L3: please change the word “dozen” with another more scientific word

We wanted to emphasize that the number of interacting molecules is high, we changed “dozen” to simply “many”.

• Introduction: in the sentence “In humans, polymorphism within the LTα …”, please define LT the first time you mention it.

 We added the definition (marked in blue, as it was also requested by the Reviewer #1)

• Material and Methods: I ask the Authors a brief comment in the main-text about the sentence: “samples from 2005 were sequenced using 150 cycles, and samples from 2016 using 300 cycles.

Number of cycles is a characteristic of the sequencing cycle on MiSeq machine. We changed the text to perhaps more often used alternative: “using 2x75 paired-end kit, and samples from 2016 were processed using 2x150bp paired-end kit”

• Material and Methods: In the sentence: “In a second step, we excluded from the model variants with MAF<0.5 and those present in fewer than 5 animals (see Table S4), as we lacked statistical power to test for their effects”, I understand the reasons of a lack of statistical power, but, in my opinion, a new mutation may be (at the beginning) present in only one animal and with a MAF<0.5. Please comment this fact in the main-text.

Yes, we are aware that this is an usual problem with estimating the effect of rare alleles, and obviously a beneficial mutation will initially be present in single (or few) animals. Practically, it is impossible to test for effect of alleles of MAF <5% when the relative risk of disease is lower that 1.5 (Foulkes 2008), and such strong risks can be rarely observed in wild systems. Although new methods to deal with rare variants have been developed (eg. SKAT, Wu et al.2011, Am J Hum Genet 89), they are not suitable to deal with response variable of Poisson distribution such as parasite counts. We added this comment to the text. 

Supplementary Tables 2: the Authors indicated that some primers contain degenerate nucleotides (such as W, Y, etc.); I ask the Authors to add a brief comment and an explanation of this in the main-text

We do not think it is necessary to comment, as degenerate primers, in particular those with a single degenerate site, are commonly used. Nonetheless, we added the word “degenerate” to the passage where we refer to Table S2. 

Response to the Reviewer #3:

I do feel that the introduction could be strengthened by giving a bit more background to the study system and stating the rationale for looking at these particular cytokine genes in this host-parasite system. 

 We expanded the Introduction adding a passage describing the study system. We also added the hypothesis and the goals. 

In addition, I think the methods could be a bit clearer on the details of the study system and statistical analyses. 

 Following the Reviewers remarks, we clarified and completed the Methods with a focus on statistical approach. 

There are also some grammatical errors and typos, which I have tried to point out as best as I can despite the lack of line numbers. I go through these points in more detail, along with other comments that I hope will improve the clarity of the manuscript, below:

We greatly appreciate the effort and we apologized for the lack of line numbering that we somehow missed in the submission process. The minor grammatical errors and typos were corrected in the text but for simplicity, we did not mark them in in colour. The text was sent for professional English editing service.

Abstract: would be nice to add in the sample size into the abstract

We agree that sample size should be provided but in the number of genotyped animals differed between studied genes. Yet, it takes at least 2 sentences to explain which may be too complex for an abstract. 

Abstract, line 9: I think this should read ‘one of them’ rather than ‘was of them’

 Yes, this was an editing mistake, the sentence was corrected (correction in blue, as it was also pointed out by the Reviewer 1)

Abstract, line 10: would be nice to add in the type of selection observed

 Added

Abstract, final sentence: it appears that there are more associations with coding (exonic) variants that affect parasite infection/load than intronic variants?

We clarified the sentence so that it accurately describes the results present in the reviewed version of the paper

Introduction, paragraph 2, line 6 and 11: might need to specify that selection was found in the genes (line 6) and variation was genetic variants in interleukin genes (line 11)

Thank you for pointing out this too-often used mental shortcut, of course the selection acts on genes, not on the proteins they code. We clarified this in the text. 

Introduction, paragraph 2, line 12: ‘A’ single study. Might need to specify whether this has been investigated and not found, or if only one study has investigated non-coding regions

 To clarify, we changed “A study” to “ Only one study focusing on non-coding variance in free- living animal”.

Introduction, paragraph 2: in the next paragraph you explain what LTα is but it would be better to put this here on the first mention of this cytokine gene.

 We explained the abbreviation here (as requested by the other Reviewers) but we opt to explain the function later, where we introduce LTα along with other two cytokines studied in the paper.

Introduction: I believe that the introduction would benefit from bringing in the study system and describing the types of pathogens that are present in the two populations of bank voles. What pathogens cause more morbidity or mortality in this system? It would then be nice to use this information to link into why you are studying these particular cytokine genes so that it feels more hypothesis driven. This would also help you to develop key expectations of the types of genes that might be important – from previous work for example by Turner et al in the field voles – and also what types of variants might be important (e.g. synonymous vs non-synonymous and exonic vs intronic).

We re-arranged the Introduction so that it now consists the second paragraph formerly in Methods describing the study system. We added a brief info about the fitness consequences of infections (although there is surprisingly few studies on that topic in wild systems). We also better formulated the hypothesis and we clarified aims. 

Materials and methods, section 2.1: It would be important to add here the sample size of voles in the two years at each site into the main text (rather than a supplementary table). Later, post QC of SNPs it would be good to state from X number of individuals X were used in the final analyses.

 We moved table S1 to the main text (now Table 1), and we amended it so that it shows number of samples collected / sequenced and number of samples included in the final analyses.

Materials and methods, section 2.1: Are any of the individuals in these populations related? Did you control for this at all?

We did not controlled for relatedness in the current paper. Yet, samples collected in 2005 and used in the current paper have been genotyped in 7 microsatellite loci and those results are reported in Kloch et al. 2010 and were later used in Kloch et al. 2018. 

 In the 2018 paper, we fitted first and second principal components (PC1 and PC2) of the relatedness matrix in GLM models analysing links between genetic variants in TLR genes and susceptibility to infections. In any model these two variables were significant and their effect sizes were low. 

 In the current paper we did not include relatedness data as we lack microsatellite analysis for samples collected in 2016. However, based on the 2018 results we are pretty confident that relatedness did not affect the outcome of the current study. 

Materials and methods, section 2.1: A brief discussion of the differences in parasite burden between the two sites would be good to have here (or in the introduction) so that this manuscript can stand alone.

We added a description of the study system to the Introduction and we explained the differences between sites. Generally, the main difference is site-specific occurrence of H. mixtum and H. glareoli. The rest of GI nemetodes is observed in all sites. Their abundance vary from year to year but there is no consistent temporal pattern.

Materials and methods, section 2.1: In the introduction and discussion it would be important to introduce and compare respectively the associations you find here with those in the MHC and TLR genes in the same study system – are patterns of synonymous vs non-synonymous and exonic vs intronic similar? Are effect sizes larger in the MHC or TLR genes than in the cytokine genes

Both previous studies (MHC and TLR) included haplotypes, not individual SNPs, and comprised only exonic sequences. Since the previous studied focused on functional difference between variants, in the previous studies we focused on haplotypes rather than single nucleotide substitutions as in the current paper. In the MHC study we analysed whole haplotypes of the exon2 of MHC DRB, and in the TLR study we analysed amino-acid haplotypes. Hypothetically, we can extract the synonymous/non-synonymous data from our archives but this would take much more time than available for the review. Nonetheless, we added a passage referring to our previous results.

Materials and methods, page 5, first paragraph: why were a different number of cycles used?

Between first and second sequencing the prices and availability of Illumina kits changed, so in the second run we were able to afford longer reads (The number of cycles equals to the read length).

Materials and methods, page 5, second paragraph: it would help to give a version and reference for both the bank vole genome and mouse genome used (for orthologues).

We added the reference to the mouse sequences (in blue as also requested by Reviewer #1). For bank vole reference, we used draft genome with BioProject accession no. PRJNA290429 which was described in the third paragraph of the section 2.2.

Materials and methods, page 5, second paragraph: it would help the reader if you explained what the abbreviations are that are used as your QC criteria in vcffilter.

We added a description, and now the passage reads: “The results were filtered […] with following conservative criteria: remove low quality calls (QUAL/AO > 10), remove loci with low read depth (DP > 10), remove alleles present in only one strand (SAF > 0 & SAR > 0), remove alleles that are only observed by reads placed to the left or right (RPR > 0 & RPL > 0).

Materials and methods, page 5, second paragraph, final sentence: would be nice to add in the number of genes and corresponding chromosomes

We are sorry but we don’t quite understand this comment. We studied only 3 genes. Did the Reviewer mean number of haplotypes/alleles found in our study or number of SNPs per gene? We added a short summary of the genetic variance in the studied loci in section 3.1, as requested also by Reviewer 1.

We cannot provide chromosomes, as the chromosome-resolved bank vole genome has not been released yet and the location of the studied genes on chromosomes in vole genome is unknown.

Materials and methods, page 5, section 2.3, first paragraph: I don’t really understand the last sentence about minimising false positives– could you clarify?

We agree that this sentence was unclear. We meant that it is hard to provide statistically robust analysis when individual with a rare allele has a rare parasite. Increased sample size might help but we could not sample more animals due to ethical reasons. Thus, we applied several filters removing variables that were too few to produce reliable results. We rewrote this passage to make our point clear. 

Materials and methods, page 5, section 2.3, second paragraph: do you mean you removed SNPs with a genotyping rate <100% and any individuals that had any missing SNP genotypes? Please give final sample sizes after QC

 We clarified that including number of samples before and after filtering in the Table 1. 

Materials and methods, page 5, section 2.3, second paragraph: How many SNPs deviated from HWE? This might be expected if selection is occurring on these loci.

We agree that selection could affect HWE but there can be also other reasons for disequilibrium. We decided to remove loci not in HWE following recommendations for association analysis (PLINK manual for example). This step is advised as such variants may result from genotyping errors. Due to HWE, we removed 8 SNPs in TNF, 3 in LTa, and 0 in IFN. In tests of selection, we used all variants, regardless of HWE.

Materials and methods, page 6, first paragraph: I think there is a typo – surely minor allele frequency must be <0.5?

 Yes, this was a typo, we meant 0.05. 

Materials and methods, page 6, first paragraph: rather than put the pathogens in a supplementary table it would be important to specifically list each pathogen looked at and ideally the % prevalence. Histograms of parasite load would be nice and should go in supplement. I think this is really important because it is not immediately obvious to the reader how many pathogens you are looking at and therefore it is hard to put the results into context.

We moved the Table S5 that shows prevalence in each sample set (i.e. in samples genotyped in each locus) to the main text, now it’s Table 2. We also added a passage summarizign this data to. Graphs of parasite loads are in Figure S1.

Materials and methods, page 6, second paragraph: some details of the models are missing including the type of link, how significance of fixed effects was determined, how post-hoc testing was done between SNP genotypes, which package was used for models etc. How did you control for overdispersion (and presumably zero-inflation) in the poisson model? Or did the model fit without other means of controlling for overdispersion? Did every pathogen abundance/load model suitably fit a poisson model? Did you try running these in a hurdle model which should combine the two models you mention (parasite infection y/n and parasite load)? Did you test for any interactions between SNPs and non-genetic sources such as sex or site?

 We wished to make this section concise but we admit that indeed several details are missing, and we completed section 2.3. Parasite presence/absence was modelled using binomial distribution with logit link function, and abundance was fitted using Poisson distribution and log link function. To control for overdispersion, we used quasi-Poisson errors implemented in glm function in R library {stat}. The significance of terms was determined using LR type III tests. We tested for interactions but we lacked df to do that properly i.e. not all combinations of alleles and non-genetic factors were present in the data resulting in a poor fit of the models. 

 As explained in the letter to the Editor, before submitting the original paper we examined several statistical approaches (eg. mixed models with non-genetic data as fixed terms, stepwise-simplified models, models with negative binomial distribution for count data instead of Poisson etc.) to conclude that all produced similar results. They all indicated the same set of SNPs to be significant, although the exact p-values varied a bit between models. We present the models with simplest structure as in more complicated models we encountered problems related to data structure such as separation or singularity, and the fact that the same SNPs were significant regardless the model structure makes us confident that the presented effects are valid.

Materials and methods, page 6, section 2.4: it would be useful to the reader if you could explain how these codon-based tests identify sites under selection, even briefly.

 The added a brief explanation on what do the tests do. We also clarified results of those tests in section 3.2

Results: I think it would be useful to explicitly state each of the null results (for both models and each pathogen) you found too rather than just the significant results. It would also be good to state X number of SNPs out of X total SNPs in a gene were associated with a trait – as you did with TNF in the first sentence. It would also be good to state whether SNP effects were additive, and to compare the effects of different SNPs (on the same parasite) in the discussion.

 We rewrote this section of results following the suggestion.

Results: There is some inconsistency in how you explain the results. For some SNPs you give the average number of worms in each genotype class but not for others – it would be good to provide this for all. Also it would be good to have errors around these values and to explain how these were calculated in the methods – is it from the raw data or post hoc testing controlling for other non-genetic variables in the model?

 We clarified the description of the results and in Fig. 1

Results: I would not discuss a result that did not meet the p value after correction for multiple testing.

 We removed this part of the discussion

Results: It would be nice to have full results from each model (model estimates, errors and p values for each SNP and for non-genetic variables too and post hoc testing between genotypes) reported somewhere, even if in the supplement. You could also compare the effects of non-genetic sources to SNPs in each model

 Full results from each model are in Table S5. We added parameter estimates and effect sizes. 

Results, page 7, second paragraph: what does average high mean?

 Clearly it is some editing typo, in the reviewed version we rewrote the sentence.

Figure 1: Explain bar chart colours and outliers and labels. A stacked bar chart might be a better option.

 In the caption we explained colours, and defined outliners. Also we changed bars to stacked. 

Discussion: it would be nice if the discussion was reformatted to follow the same format as the results – starting with SNPs associated with infection or parasite load, followed by discussion of selection on these variants

 We reformatted as requested. The discussion is now divided into three sub-sections (the extra one that does not follow the Result pattern is about the role of non-coding variants).

Discussion, page 8, line 3: I would rephrase to ‘genetic variation in cytokine genes also plays’ rather than ‘cytokine variance’ as this sounds like you are actually measuring cytokine levels

 We rephrased the sentence 

Discussion, page 8, paragraph 2: there are a lot of ‘what’s in this paragraph that should be ‘which’

 We corrected that.

Discussion, page 8, paragraph 2: I wouldn’t comment on the effect on Bartonella sp. as I think it was non significant after correction for multiple testing

 After applying Bonferroni correction this result turn non-significant and we removed Bartonella from discussion.

Discussion, page 8, paragraph 2: following on from the final sentence it would be nice to have some insight into how these pathogens affect morbidity and mortality in this species or a similar species

 We added a paragraph describing effect of nematode infections on host fitness in rodents. 

Discussion, page 8, paragraph 3: This is a nice discussion of the role of synonymous mutations. Thank you 

 Thank you :)

Discussion: some more discussion of other wild systems (e.g. Turner et al’s work in the field voles) and comparisons to effects of the MHC and TLR variants previously investigated in this system would strengthen the manuscript. It would also be nice to add whether variants in IFNβ1 and LTα (or orthologues) have previously been implicated in susceptibility to parasitic diseases in other species.

 We elaborated this part of the Discussion.

Discussion, page 9: the conclusion feels a bit rushed. While there is an intronic and a synonymous SNP, it appears that more associations with exonic and non-synonymous SNPs and parasite infection/load – so can you say that intronic and synonymous variants play a (more) important role? Or are you just saying that they are also present and should not be ruled out?

In the Conclusion, we added a brief summary of the key findings. At the moment, we cannot say how important is their role but our findings clearly suggest that they should not be disregarded in future studies. We clarified our point.

Discussion: some discussion of the advantages and disadvantages of candidate gene studies would be helpful to the reader

Having dealing with candidate-genes vs. genome-wide scans in our work, we see this topic as too broad to be included in the Discussion. To discuss the pros and cons we would need to get into technical details and we think that this is rather off-topic to the main theme of the paper.

---

## [Decision Letter · Decision Letter 1]

1 Sep 2022

PONE-D-21-29710R1Cytokine gene polymorphism and parasite susceptibility in free-living rodents: importance of non-coding variantsPLOS ONE

Dear Dr. Kloch,

Thank you for submitting your manuscript to PLOS ONE. After careful consideration, we feel that it has merit but does not fully meet PLOS ONE’s publication criteria as it currently stands. Therefore, we invite you to submit a revised version of the manuscript that addresses the points raised during the review process.

All reviewers agree that your manuscript is interesting but two of them consider that it still needs of major revisions before a final acceptance.

Two importants aspects would be particularly corrected:

- the first one concerns the risks of false positives and the interest to use multiple testing corrections. One reviewer would be happy if you could add some text to the discussion to indicate the caveats of this approach and the follow up studies needed to confirm any associations which are observed in this study.

- the second one concerns positive selection analyses. Positive selection analysis cannot be performed on exon-intron sequences and authors need to redo the analysis only on CDS regions and re-interpret results where necessary.

Different other comments are given in the new reviews. I suggest you to consider all of them, as they will improve the quality of your study.

We look forward to receiving your revised manuscript.

Kind regards,

Johan R. Michaux

Academic Editor

PLOS ONE

Reviewers' comments:

Reviewer's Responses to Questions

**Comments to the Author**

1. If the authors have adequately addressed your comments raised in a previous round of review and you feel that this manuscript is now acceptable for publication, you may indicate that here to bypass the “Comments to the Author” section, enter your conflict of interest statement in the “Confidential to Editor” section, and submit your "Accept" recommendation.

Reviewer #1: (No Response)

Reviewer #2: All comments have been addressed

Reviewer #3: (No Response)

2. Is the manuscript technically sound, and do the data support the conclusions?

Reviewer #1: Partly

Reviewer #2: Yes

Reviewer #3: Partly

3. Has the statistical analysis been performed appropriately and rigorously? 

Reviewer #1: Yes

Reviewer #2: Yes

Reviewer #3: Yes

4. Have the authors made all data underlying the findings in their manuscript fully available?

Reviewer #1: Yes

Reviewer #2: Yes

Reviewer #3: Yes

5. Is the manuscript presented in an intelligible fashion and written in standard English?

Reviewer #1: Yes

Reviewer #2: Yes

Reviewer #3: Yes

6. Review Comments to the Author

Reviewer #1: I appreciate that authors have carefully addressed my comments and significantly improved the quality of the manuscript. I now have one major comment concerning positive selection analysis that I unfortunately missed previously. Positive selection analysis cannot be performed on exon-intron sequences and authors need to redo the analysis only on CDS regions and re-interpret results where necessary. Please, find some other comments below.

1) L49, Parasites may affect the genetic variation of their hosts through three main mechanisms.

There also some other „main“ mechanisms, such positive or purifying selection. Better opening sentence for BS is needed.

2) L127, All three sites are on public ground managed by the Polish State Forests 128 and no specific permission to access the land was required.

This sentence ca be omitted.

3) L171, A fraction of SNPs (5 of 36 in TNF, and 6 of 18 in LTα) could 172 not be phased by FreeBayes.

To avoid confusion, better to use „reconstructed“ over „phased“. Also, better to write, a fraction of alleles…

4) Table 1 can be better placed in SI as it shows only supportive information.

5) L233, We overcome this drawback by using two additional 234 tests (MEME and FUBAR) to test for episodic selection.

FUBAR detect only pervasive selection. Please, correct the statement.

5) L238-239 should be moved to the methods.

6) L346-349. This is redundant sentence as mechanism of NFSD is already well-explained in the introduction.

7) L357, Given the absence of revealed recombination, those results must be interpreted with caution. Without direct functional testing using mouse models, the revealed association between parasite load and particular SNPs must be interpreted with caution as it can be caused by some other adjacent variation. Please, incorporate this point in the discussion.

7) L642,

It seems that authors analysed pattern of positive selection across whole exon-intron sequences. This is misleading since dN/dS based selection method are intended only for protein-coding data. Introns are under different mode of evolution. I ask authors to perform selection analyses only on protein-coding data, i.e. CDS region and change the interpretation of data where necessary. Also, please, explicitly mention in the methods that selection analysis was performed only on exonic sequences

8) Figure 2

Why do authors not provide location of sites under selection also for TNF?

Reviewer #2: (No Response)

Reviewer #3: I reviewed the earlier version of the manuscript and I am pleased to see that the majority of comments and suggestions from the reviewers have been addressed. I would like to highlight, however, that the reviewing process would have been easier if the authors had added line numbers corresponding to each of their edits for each of their responses to the editors and reviewers comments.

The manuscript reads much better following revision, and I do believe that all the sections of the manuscript have been strengthened. However, I am still slightly skeptical about the associations since population structure can lead to false positives. Ideally, population structure would be accounted for in some way, or a set of control genes would also be included to check whether the test statistic (lambda) follows the null distribution or whether it is inflated. I am aware however that the authors do not have this data, and I am also aware that there are a lot of candidate gene studies out there that do not account for population structure, and I do believe that multiple testing correction in the revision has reduced the chance of false positives. As a result, I would be happy if the authors added some text to the discussion to indicate the caveats of this approach and the follow up studies needed to confirm any associations they see.

Comments to the authors response to the editor and reviewers:

• In terms of the non-genetic sources of variation to include in the model: I would usually advocate to keep all terms in the model as a more conservative approach. However, I understand that overfitting is a problem particularly when the sample sizes are low. I would trust the authors assertion that the models were similar if they added the alternative models with all fixed effects fitted (non-genetic sources and SNPs) to the supplement. I also think it would have been an easier approach to test – for each parasite and each parasite model (absence or parasite load) – the fixed effects rather than checking for each parasite model/gene subset as they could be significant in one and not the other just due to small changes in sample size.

• I do feel that the results are strengthened since applying the Bonferonni correction. I think ideally the number of tests would be - the number of parasites tested (5?) x number of parasite models (2) x number of genes (as SNPs within a gene might not be independent?) or SNPs. I would be interested to see what the other reviewers think of the criteria of 10 tests because I understand that my suggestion would be very very conservative. I would also like to note that given the current criteria the association between LTa 322 and A. tianjinensis is technically not significant (p=0.0054) and reference to this association should be removed.

• I would add the data availability statement into the manuscript

• You mention that you exclude variants (genotypes?) present in fewer than 5 animals but then I am unsure why LTa 322 is included since it has no CC genotype (or had 1 animal which was removed)?

• I feel that the introduction is much improved by the addition of information about the study system. I do feel as though some hypotheses are a bit vague – as well as the statement that the authors focused on these genes as “little is known”. I did expect some more references to human GWAS which may have found SNPs associated with these genes and parasite/pathogen infections, although perhaps there are none. Here it would be better to specifically indicate the parasite species studied rather than e.g. ‘blood microparasites’.

• Page 2, line 108 – parasite load of what species?

• The statistical methods are a lot clearer now, thank you.

• I appreciate the re-writing of the conclusion when I queried how important the non-coding variants are – I wonder also if this part of the title could be removed as you did find more coding variants.

• I appreciate that the authors brought in the table of parasite species (Table 2) into the main manuscript in response to my query that I wanted the parasite species tested mentioned specifically in the text. However, I do feel like there is a lot of information in this table for the main manuscript. I would really prefer if the authors just listed the species studied directly in the text (referring to the type of parasite they are too) and put this table back in the Supp. I suggest this because it is much easier for readers to work out which species are looked at than having to look up the table. I also found the table difficult to interpret because I was unclear if parasite species were included if it was 20-80% infections across all gene subsets or only if in any – and if the latter – did you only run models where prevalence was > 20% for a given parasite/ gene combination (i.e. was H. glareoli only run with IFN variants and not the other two genes?). Therefore, I feel like it would be helpful to add the parasite species tested directly to the text as it was not immediately clear to me. If the authors elected to keep the table in the main manuscript you could remove the numbers infected and non-infected (as you could work this out from Table 1 from the % prevalence) to make it simpler. Either way I would highlight the species you tested in bold to help the reader.

Other comments

• I would add the parasite species tested in the abstract. I imagine most people would be looking specifically for genetic variants associated with a given parasite (or closely related parasite) so this would help their search.

• Page 3, line 74 – “non-coding variance” = non-coding variants

• Add how species abundance (not just prevalence) was measured to the methods.

• Add season/month of sample collection to methods.

• Methods - are quasi-binomial and quasi-poisson not two different types of model?

• Results – I would add null results for each parasite/gene combination.

• Table 3 – I am glad the authors have put the full model results in the main manuscript. However, the formatting is not as nice as the old table and I am unclear why there are lines around host body mass. I would recommend adding a reference to the supp table with the other (non-sig) results to the legend. There is a reference to padj which isn’t reported in the table.

• I think it would be nice to add to the summary at the start of the discussion which genes and which parasite species tested were not significant.

• Page 10, line 291 – change individuals to worms

• Page 10, line 298 – SNP variance = SNP variants

• Page 10, line 305 – generation *of* T cells

• Page 10, line 310 – but you also investigated the association between TNF and other non-nematode infections?

• Page 11, line 330 – strengthened

7. PLOS authors have the option to publish the peer review history of their article (what does this mean?). If published, this will include your full peer review and any attached files.

Reviewer #1: No

Reviewer #2: No

Reviewer #3: No

---

## [Author Response · Author response to Decision Letter 1]

31 Oct 2022

Reviewer #1:

Changes in text following Reviewer’s #1 suggestions are marked in blue.

We refer to line numbers in “track changes” version of the revised manuscript.

I appreciate that authors have carefully addressed my comments and significantly improved the quality of the manuscript.

Thank you. We are glad that we managed to satisfy the Reviewer remarks.

I now have one major comment concerning positive selection analysis that I unfortunately missed previously. Positive selection analysis cannot be performed on exon-intron sequences and authors need to redo the analysis only on CDS regions and re-interpret results where necessary. 

Thank you for pointing out this important mistake. We calculated the tests again using only exonic fragments, changing relevant parts in Methods and Results (lines 290-297). (There were no changes in IFNb1, as it consisted of exonic part only). 

1) L49, Parasites may affect the genetic variation of their hosts through three main mechanisms.

There also some other „main“ mechanisms, such positive or purifying selection. Better opening sentence for BS is needed.

We agree with the remark, although summarizing such a complex idea in a single sentence is challenging. Nonetheless, we attempted to better introduce BS (lines 48-50).

2) L127, All three sites are on public ground managed by the Polish State Forests 128 and no specific permission to access the land was required. This sentence ca be omitted.

This sentence was added upon request of the Editor: „In your Methods section, please provide additional information regarding the permits you obtained for the work. Please ensure you have included the full name of the authority that approved the field site access and, if no permits were required, a brief statement explaining why.”

3) L171, A fraction of SNPs (5 of 36 in TNF, and 6 of 18 in LTα) could 172 not be phased by FreeBayes. To avoid confusion, better to use „reconstructed“ over „phased“. Also, better to write, a fraction of alleles…

By using the term „a fraction of SNPs” we wanted to precisely depict the situation. In the vcf file resulting from our pipeline most of the SNPs were phased (i.e. assigned to given DNA strand) based on their positions on sequenced Illumina fragments (reads). If two SNPs are located in the same read, the FreeBayes assumes that they originated from the same DNA strand. This method does not work for SNPs that are separated from other SNPs by more than a read length and thus never occurr in a read with any other SNP. Here we had to use PHASE algorithm and assigned them computationally to given DNA strands.

We clarified the sentence, so now it reads “a fraction of SNPs (5 of 36 in TNF, and 6 of 18 in LTα) could not be phased by FreeBayes; these were computationally assigned to DNA strands using PHASE (Stephens et al. 2001). Reconstructed alleles were converted to fasta format ..” (lines 188-190).

4) Table 1 can be better placed in SI as it shows only supportive information.

The table was initially in the SI but was placed in the main text upon request of the Reviewer 3: „Materials and methods, section 2.1: It would be important to add here the sample size of voles in the two years at each site into the main text (rather than a supplementary table).” We can move it back to the SI if both Reviewers (#1 and #3) agree.

5) L233, We overcome this drawback by using two additional tests (MEME and FUBAR) to test for episodic selection. FUBAR detect only pervasive selection. Please, correct the statement.

We agree with this remark. After consideration, we decided to remove two final sentences of this paragraph, as the principles of all the tests are described above. We kept the information that FEL assumes constant selection pressure across phylogeny. 

5) L238-239 should be moved to the methods.

This passage was added upon request of the Reviewer #1 in the first round of reviews: “ *P6, 3.1. Cytokine polymorphisms and susceptibility to infections. I would add some general summary of polymorphism revealed, ideally with a table in SI with values such as number of sequences, number of unique nucleotide alleles, number of variable sites, number of substitutions to see how much are cytokines variable.” Nontheless, in the current review, we moved it to the end of section 2.2, lines 196-199.

6) L346-349. This is redundant sentence as mechanism of NFSD is already well-explained in the introduction.

This was added upon request of Reviewer #1: „P8, L7 Please, clarify what is meant by repeated rounds of positive selection interspersed with purifying selection.”. However, as we the explanation of NFSD was also added to the introduction, we removed this redundant fragment. 

7) L357, Given the absence of revealed recombination, those results must be interpreted with caution. Without direct functional testing using mouse models, the revealed association between parasite load and particular SNPs must be interpreted with caution as it can be caused by some other adjacent variation. Please, incorporate this point in the discussion.

We agree. We added this point to the discussion, lines 373-376.

7) L642,

It seems that authors analysed pattern of positive selection across whole exon-intron sequences. This is misleading since dN/dS based selection method are intended only for protein-coding data. Introns are under different mode of evolution. I ask authors to perform selection analyses only on protein-coding data, i.e. CDS region and change the interpretation of data where necessary. Also, please, explicitly mention in the methods that selection analysis was performed only on exonic sequences

Thank you for pointing out this mistake. We have now recalculated site-selection tests using exonic sequences only in LTa and TNF (INFb1 did not contain introns). We changed relevant parts of Methods (l. 226) and Results (section 3.2), including Fig 2.

8) Figure 2

Why do authors not provide location of sites under selection also for TNF?

Initially in Fig 2 we included only the genes where we find significant associations of given SNPs with parasite load. We agree that this may be confusing so we added TNF to the Figure.

Reviewer #3:

Changes in text following Reviewer’s #2 suggestions are marked in green.

We refer to line numbers in “track changes” version of the revised manuscript.

I reviewed the earlier version of the manuscript and I am pleased to see that the majority of coments and suggestions from the reviewers have been addressed. I would like to highlight, however, that the reviewing process would have been easier if the authors had added line numbers corresponding to each of their edits for each of their responses to the editors and reviewers comments.

Thank you for appreciation. In this review, we add line numbers corresponding to the edits.

However, I am still slightly skeptical about the associations since population structure can lead to false positives. Ideally, population structure would be accounted for in some way, or a set of control genes would also be included to check whether the test statistic (lambda) follows the null distribution or whether it is inflated. I am aware however that the authors do not have this data, and I am also aware that there are a lot of candidate gene studies out there that do not account for population structure, and I do believe that multiple testing correction in the revision has reduced the chance of false positives. As a result, I would be happy if the authors added some text to the discussion to indicate the caveats of this approach and the follow up studies needed to confirm any associations they see.

We are really thankful for understanding that some factors are difficult (or impossible) to control in the field data. We did our best to make our results as robust as possible. We added a paragraph discussing caveats of our study design in a final part of the section 4.1. 

In terms of the non-genetic sources of variation to include in the model: I would usually advocate to keep all terms in the model as a more conservative approach. However, I understand that overfitting is a problem particularly when the sample sizes are low. I would trust the authors assertion that the models were similar if they added the alternative models with all fixed effects fitted (non-genetic sources and SNPs) to the supplement. I also think it would have been an easier approach to test – for each parasite and each parasite model (absence or parasite load) – the fixed effects rather than checking for each parasite model/gene subset as they could be significant in one and not the other just due to small changes in sample size.

We added models with all genetic terms in Table S7. Please note, that for IFN there is no factor “year”, as in this gene we only genotyped samples from 2005. Similarly, there is no “year” for Cryptosporidium which was analysed only in samples from 2005. 

I do feel that the results are strengthened since applying the Bonferonni correction. I think ideally the number of tests would be - the number of parasites tested (5?) x number of parasite models (2) x number of genes (as SNPs within a gene might not be independent?) or SNPs. I would be interested to see what the other reviewers think of the criteria of 10 tests because I understand that my suggestion would be very very conservative. I would also like to note that given the current criteria the association between LTa 322 and A. tianjinensis is technically not significant (p=0.0054) and reference to this association should be removed.

Thank you for rising this point. Prior the publication, we discussed between us what should we consider multiple comparisons. For instance, we tested for linkage between SNPs within a locus as described in the first paragraph of the section 2.3. so technically they should be independent, yet statistically the more explanatory variables, the higher probability that any of them is significant by chance. Thus, we used the criterion of 10, as this was the number of explanatory variables whose effect we were interested in (the non-genetic terms were included only to control for their contribution to the observed variance in parasite load). 

 Probably it has to be clarified that due to a fact that prevalence was <20% in some combinations of loci and parasites, in total we run 23 models as shown in the table below. (This information was added in line 220). If we consider all combination of parasite x type of test x SNP as multiple comparisons, this gives in total 74 “comparisons” resulting in p-value 0.05 /74 = 0.00068. This super-conservative criterion still shows significant association between IFNβ 105 and prevalence with H. glareoli. 

Table. Combinations of parasites / genes tested in the GLM models. In total we run 23 tests. When each SNP is considered, this gives 74 comparisons. 

Presence / absence

16 models

Abundance

7 models

parasite / pathogen

TNF

LTα

IFNβ1

TNF

LTα

IFNβ1

Aspiculuris tianjensis

1 SNP

7 SNP

2 SNP

1 SNP

7 SNP

2 SNP

Heligmosomum mixtum

1 SNP

7 SNP

2 SNP

1 SNP

7 SNP

2 SNP

Heligmosomoides glareoli

-

-

2 SNP

-

-

2 SNP

Cryptosporidium sp.

1 SNP

7 SNP

2 SNP

-

-

-

Babesia microti

1 SNP

7 SNP

2 SNP

-

-

-

Bartonella sp.

1 SNP

7 SNP

2 SNP

-

-

-

 However, we doubt if models of abundance and presence/absence should be considered as multiple comparisons, as biologically each of them involves different hypothesis. Resistance against parasites involve different immunological mechanisms (i.e. preventing a pathogen from colonizing the hosts) than dealing with the number of pathogens once the infection happened. If we agree with this assumption, the threshold p-val for presence/absence tests is 0.05 / 52 (10 SNPs tested in 5 parasites + 2 SNPs tested for H. mixtum) = 0.00096 and for abundance this value is 0.05 / 22 (10 SNPs for two parasites plus 2 SNPs tested for H. mixtum) = 0.0023. Again, with those thresholds our results remain valid. 

 Finally, we think that corrections for multiple comparisons can be applied only to explanatory variables, as we are interested in minimizing risk of false positives. Models with different parasites species as response variables do not fulfil this criterion, as finding an association between genetic marker and susceptibility to disease A is independent from susceptibility to disease B - at least statistically, if we don’t assume any underlying biological mechanisms that may link these two diseases. Thus, we simply corrected for the number of loci tested. Such an approach is recommended in human GWAS studies, for instance in PLINK manual. 

We agree that the association between LTa 322 and A. tianjinensis is weak and should not be reported so we adjusted Results and Discussion. 

I would add the data availability statement into the manuscript

Initially, it was included in the PloS submission forms but it seems it did not appear in the manuscript. We added the statement to the main text. lines 451-455

You mention that you exclude variants (genotypes?) present in fewer than 5 animals but then I am unsure why LTa 322 is included since it has no CC genotype (or had 1 animal which was removed)?

We removed this one individual with the genotype CC from the analysis so that we still can test for the effect of LTa322. 

I feel that the introduction is much improved by the addition of information about the study system. I do feel as though some hypotheses are a bit vague – as well as the statement that the authors focused on these genes as “little is known”. I did expect some more references to human GWAS which may have found SNPs associated with these genes and parasite/pathogen infections, although perhaps there are none. Here it would be better to specifically indicate the parasite species studied rather than e.g. ‘blood microparasites’.

We added some text to this paragraph to make it more specific. Although we did not intend to cite many human GWAS studies, as they usually have much power and different design compared to wild systems, we included a reference to a review by Khan and Qidwai 2011 (line 123). We rephrased the sentence in line 97 (“little is known”), and we reformulated the hypotheses. 

Page 2, line 108 – parasite load of what species?

We rephrased the hypothesis and we made it more specific (lines 123-125 and 129-134).

The statistical methods are a lot clearer now, thank you.

We are happy that we succeeded to make the text clearer.

I appreciate the re-writing of the conclusion when I queried how important the non-coding variants are – I wonder also if this part of the title could be removed as you did find more coding variants.

We agree to remove “non-coding variants” from the title after acceptance from the Associated Editor.

I appreciate that the authors brought in the table of parasite species (Table 2) into the main manuscript in response to my query that I wanted the parasite species tested mentioned specifically in the text. However, I do feel like there is a lot of information in this table for the main manuscript. I would really prefer if the authors just listed the species studied directly in the text (referring to the type of parasite they are too) and put this table back in the Supp. I suggest this because it is much easier for readers to work out which species are looked at than having to look up the table. I also found the table difficult to interpret because I was unclear if parasite species were included if it was 20-80% infections across all gene subsets or only if in any – and if the latter – did you only run models where prevalence was > 20% for a given parasite/ gene combination (i.e. was H. glareoli only run with IFN variants and not the other two genes?). Therefore, I feel like it would be helpful to add the parasite species tested directly to the text as it was not immediately clear to me. If the authors elected to keep the table in the main manuscript you could remove the numbers infected and non-infected (as you could work this out from Table 1 from the % prevalence) to make it simpler. Either way I would highlight the species you tested in bold to help the reader.

We put Table 2 back in the Suppl Mat (now S4). We run only tests where prevalence was >20% for given gene/parasite combination, we clarified that in the text and we summarized the models tested (lines 220-226).

I would add the parasite species tested in the abstract. I imagine most people would be looking specifically for genetic variants associated with a given parasite (or closely related parasite) so this would help their search.

Added.

Page 3, line 74 – “non-coding variance” = non-coding variants

Corrected

Add how species abundance (not just prevalence) was measured to the methods.

Worms of each species were counted. The counts were used as abundance. We added this information in the line 135. 

Add season/month of sample collection to methods.

Added. In both years it was September (line 138).

Methods - are quasi-binomial and quasi-poisson not two different types of model?

Yes, they are. The sentence may be confusing so we clarified “in abundance models we used quasi-Poisson errors and in presence models quasi-binomial errors” (line 235)

Results – I would add null results for each parasite/gene combination.

We added a suitable paragraph to the first part of the section 3.1. (lines 266-273)

Table 3 – I am glad the authors have put the full model results in the main manuscript. However, the formatting is not as nice as the old table and I am unclear why there are lines around host body mass. I would recommend adding a reference to the supp table with the other (non-sig) results to the legend. There is a reference to padj which isn’t reported in the table.

The lines were errors in editing. We corrected the caption. In the reviewed version we removed results for LTAa 322 (marginally non-significant, p=0.0052) which prompted us to reformat the table. 

I think it would be nice to add to the summary at the start of the discussion which genes and which parasite species tested were not significant.

Added (lines 304-309)

Page 10, line 291 – change individuals to worms 

Corrected

Page 10, line 298 – SNP variance = SNP variants

corrected to „variants”

Page 10, line 305 – generation *of* T cells

corrected

Page 10, line 310 – but you also investigated the association between TNF and other non-nematode infections?

Yes, we clarified this (line 337)

Page 11, line 330 – strengthened

corrected

---

## [Decision Letter · Decision Letter 2]

8 Jan 2023

Cytokine gene polymorphism and parasite susceptibility in free-living rodents: importance of non-coding variants

PONE-D-21-29710R2

Dear Dr. Kloch,

We’re pleased to inform you that your manuscript has been judged scientifically suitable for publication and will be formally accepted for publication once it meets all outstanding technical requirements. We would just like to take into account the last minor comments suggested by one of the reviewers.

Kind regards,

Johan R. Michaux

Academic Editor

PLOS ONE

Additional Editor Comments (optional):

Reviewers' comments:

Reviewer's Responses to Questions

**Comments to the Author**

1. If the authors have adequately addressed your comments raised in a previous round of review and you feel that this manuscript is now acceptable for publication, you may indicate that here to bypass the “Comments to the Author” section, enter your conflict of interest statement in the “Confidential to Editor” section, and submit your "Accept" recommendation.

Reviewer #1: (No Response)

Reviewer #2: All comments have been addressed

2. Is the manuscript technically sound, and do the data support the conclusions?

Reviewer #1: Yes

Reviewer #2: Yes

3. Has the statistical analysis been performed appropriately and rigorously? 

Reviewer #1: Yes

Reviewer #2: Yes

4. Have the authors made all data underlying the findings in their manuscript fully available?

Reviewer #1: Yes

Reviewer #2: Yes

5. Is the manuscript presented in an intelligible fashion and written in standard English?

Reviewer #1: Yes

Reviewer #2: Yes

6. Review Comments to the Author

Reviewer #1: I again appreciate that the authors carefully integrated my comments. Now I only have a few minor comments.

L30, First mentioning of MHC and TLR abbreviations should be explained here and, in the introduction.

L33, Latin name can be added if there is space

L50 Please, add "mutually not exclusive mechanisms"

L59, Please add the reference Acevedo-Whitehouse K, Cunningham AA. Is MHC enough for understanding wildlife immunogenetics? Trends Ecol Evol. 2006 Aug;21(8):433-8. doi: 10.1016/j.tree.2006.05.010. Epub 2006 Jun 9. PMID: 16764966.

L73, Mycobacterium is not a disease

L90, Please, explain TLR abbreviation

L92-93, I do not like the sentence. Better TLRs recognize...

L196-197 I do not understand why you filtered variants that are not in Hardy-Weinberg equilibrium. I would expect that strong parasite-mediated selection going on particular SNP might deviate it from HW. Please, explain it.

L350-351 Given the context, I would use instead "further studies are needed, ideally accounting for genome-wide variation" rather "further studies should functionally verify the effect of predicted SNPs."

L363-365 Please, remove the following part: “through frequency-dependent selection” as it is speculative.

L378, I suggest using "Candidate SNPs should be verified by functional in vitro testing." better than previous "To strengthen our hypothesis, further 379 studies using direct functional tests eg. using mouse models are needed.“

L672-675, Bonferroni correction

I am not a biostatistician but I agree with the other reviewer´s conservative view on Bonferroni's correction. To obtain the critical p-value, l would simply divide p = 0.05 by the final number of all performed tests/ models from the statistical second step. To be clearer for a reader, you could also number all models in the tables and when referring to their results, you could use their model numbers in the text.

L690 Please, add reference sequence ID.

Reviewer #2: All comments have been addressed by the Authors; now, in my opinion, the present paper can be accepted for publication.

7. PLOS authors have the option to publish the peer review history of their article (what does this mean?). If published, this will include your full peer review and any attached files.

Reviewer #1: No

Reviewer #2: No

---

## [Editor Report · Acceptance letter]

13 Jan 2023

PONE-D-21-29710R2 

Cytokine gene polymorphism and parasite susceptibility in free-living rodents: importance of non-coding variants 

Dear Dr. Kloch:

I'm pleased to inform you that your manuscript has been deemed suitable for publication in PLOS ONE. Congratulations! Your manuscript is now with our production department. 

Kind regards, 

on behalf of

Dr. Johan R. Michaux 

Academic Editor

PLOS ONE